# Light Field Networks: Neural Scene Representations with Single-Evaluation Rendering

**Vincent Sitzmann**[1],[*]
sitzmann@mit.edu

**Semon Rezchikov**[2],[*]
skr@math.columbia.edu

**William T. Freeman**[1],[3]
billf@mit.edu

**Joshua B. Tenenbaum**[1],[4],[5]
jbt@mit.edu

**Frédo Durand**[1]
fredo@mit.edu

[1]MIT CSAIL    [2]Columbia University    [3]IAFI    [4]MIT BCS    [5]CBMM
vsitzmann.github.io/lfns/

## Abstract

Inferring representations of 3D scenes from 2D observations is a fundamental problem of computer graphics, computer vision, and artificial intelligence. Emerging 3D-structured neural scene representations are a promising approach to 3D scene understanding. In this work, we propose a novel neural scene representation, Light Field Networks or LFNs, which represent both geometry and appearance of the underlying 3D scene in a 360-degree, four-dimensional light field parameterized via a neural network. Rendering a ray from an LFN requires only a *single* network evaluation, as opposed to hundreds of evaluations per ray for ray-marching or volumetric based renderers in 3D-structured neural scene representations. In the setting of simple scenes, we leverage meta-learning to learn a prior over LFNs that enables multi-view consistent light field reconstruction from as little as a single image observation. This results in dramatic reductions in time and memory complexity, and enables real-time rendering. The cost of storing a 360-degree light field via an LFN is two orders of magnitude lower than conventional methods such as the Lumigraph. Utilizing the analytical differentiability of neural implicit representations and a novel parameterization of light space, we further demonstrate the extraction of sparse depth maps from LFNs.

## 1 Introduction

A fundamental problem across computer graphics, computer vision, and artificial intelligence is to infer a representation of a scene's 3D shape and appearance given impoverished observations such as 2D images of the scene. Recent contributions have advanced the state of the art for this problem significantly. First, neural implicit representations have enabled efficient representation of local 3D scene properties by mapping a 3D coordinate to local properties of the 3D scene at that coordinate [1–6]. Second, differentiable neural renderers allow for the inference of these representations given only 2D image observations [3, 4]. Finally, leveraging meta-learning approaches such as hypernetworks or gradient-based meta-learning has enabled the learning of distributions of 3D scenes, and therefore reconstruction given only a single image observation [3]. This has enabled a number of applications, such as novel view synthesis [7, 3, 6], 3D reconstruction [5, 3] semantic segmentation [8, 9], and SLAM [10]. However, 3D-structured neural scene representations come with a major limitation: Their rendering is prohibitively expensive, on the order of *tens of seconds for a single* $256 \times 256$ *image* for state-of-the-art approaches. In particular, parameterizing the scene in 3D space necessitates

---

[*]These authors contributed equally to this work.

35th Conference on Neural Information Processing Systems (NeurIPS 2021).

the discovery of surfaces along camera rays during rendering. This can be solved either by encoding geometry as a level set of an occupancy or signed distance function, or via volumetric rendering, which solves an alpha-compositing problem along each ray. Either approach, however, requires tens or even hundreds of evaluations of the 3D neural scene representation in order to render a single camera ray.

We propose a novel neural scene representation, dubbed Light Field Networks or LFNs. Instead of encoding a scene in 3D space, Light Field Networks encode a scene by directly mapping an oriented camera ray in the four dimensional space of light rays to the radiance observed by that ray. This obviates the need to query opacity and RGB at 3D locations along a ray or to ray-march towards the level set of a signed distance function, speeding up rendering by *three orders of magnitude* compared to volumetric methods. In addition to directly encoding appearance, we demonstrate that LFNs encode information about scene geometry in their derivatives. Utilizing the unique flexibility of neural field representations, we introduce the use of Plücker coordinates to parameterize 360-degree light fields, which allow for storage of a-priori unbounded scenes and admit a simple expression for the depth as an analytical function of an LFN. Using this relationship, we demonstrate the computation of geometry in the form of sparse depth maps. While 3D-structured neural scene representations are multi-view consistent *by design*, parameterizing a scene in light space does not come with this guarantee: the additional degree of freedom enables rays that view the same 3D point to change appearance across viewpoints. For the setting of simple scenes, we demonstrate that this challenge can be overcome by learning a prior over 4D light fields in a meta-learning framework. We benchmark with current state-of-the-art approaches for single-shot novel view synthesis, and demonstrate that LFNs compare favorably with globally conditioned 3D-structured representations, while accelerating rendering and reducing memory consumption by orders of magnitude.

In summary, we make the following contributions:

1. We propose Light Field Networks (LFNs), a novel neural scene representation that directly parameterizes the light field of a 3D scene via a neural network, enabling real-time rendering and vast reduction in memory utilization.

2. We demonstrate that we may leverage 6-dimensional Plücker coordinates as a parameterization of light fields, despite their apparent overparameterization of the 4D space of rays, thereby enabling continuous, 360-degree light fields.

3. By embedding LFNs in a meta-learning framework, we demonstrate light field reconstruction and novel view synthesis of simple scenes from sparse 2D image supervision only.

4. We demonstrate that inferred LFNs encode both appearance and geometry of the underlying 3D scenes by extracting sparse depth maps from the derivatives of LFNs, leveraging their analytical differentiability.

**Scope.**  The proposed method is currently constrained to the reconstruction of simple scenes, such as single objects and simple room-scale scenes, in line with recent work on learning generative models in this regime [3, 11].

## 2 Related Work

**Neural Scene Representations and Neural Rendering.**  A large body of work addresses the question of inferring feature representations of 3D scenes useful to downstream tasks across graphics, vision, and machine learning. Models without 3D structure suffer from poor data efficiency [12, 13]. Voxel grids [14–20] offer 3D structure, but scale poorly with spatial resolution. Inspired by neural implicit representations of 3D geometry [1, 2], recent work has proposed to encode properties of 3D scenes as *neural fields* (also implicit- or coordinate-based representations, see [21] for an overview), neural networks that map 3D coordinates to local properties of the 3D scene at these coordinates. Using differentiable rendering, these models can be learned from image observations only [3, 4, 22, 11]. Reconstruction from sparse observations can be achieved by learning priors over the space of neural fields [3, 5, 11, 23–25] or by conditioning of the neural field on local features [6, 26, 27]. Differentiable rendering of such 3D-structured neural scene representations is exceptionally computationally intensive, requiring hundreds of evaluations of the neural representation per ray, with tens of thousands to millions of rays per image. Some recent work seeks to accelerate test-time rendering, but either does not admit generalization [28–30], or does not alleviate the cost of

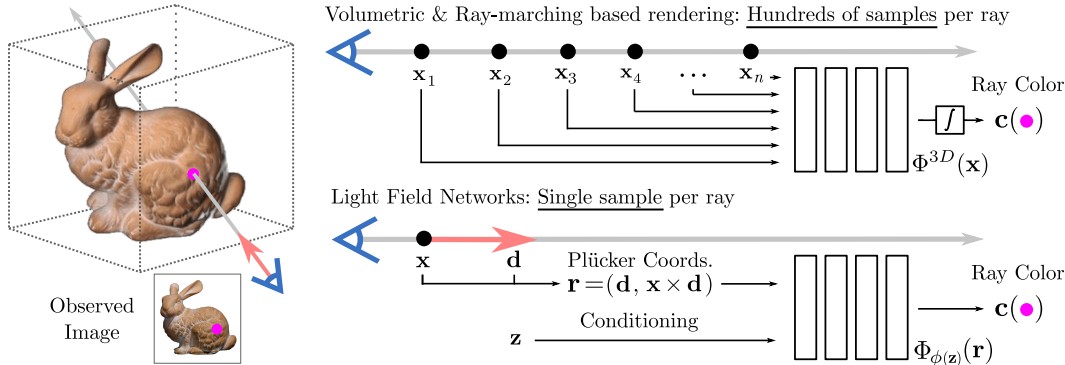

Figure 1: **Overview.** We propose Light Field Networks or LFNs, which encode the full 360-degree light field of a 3D scene in the weights of a fully connected neural network $\Phi_\phi$ (with weights $\phi$ conditioned on a latent code $\mathbf{z}$) that maps an oriented ray $\mathbf{r}$ to the radiance $\mathbf{c}$ observed by that ray. Rendering an LFN $\Phi_\phi$ only requires evaluating the underlying MLP once per ray, in contrast to 3D-structured neural scene representations $\Phi_{3D}$ such as SRNs [3], NeRF [4], or DVR [5] that require hundreds of evaluations per ray. We leverage meta-learning to learn a multi-view consistent space of LFNs. Once trained, this enables reconstruction of a 360-degree light field and subsequent real-time novel view synthesis of simple scenes from only a single observation.

rendering at training/inference time [31–33]. With Light Field Networks, we propose to leverage 360-degree light fields as neural scene representations. We introduce a novel neural field parameterization of 360-degree light fields, infer light fields via meta-learning from as few as a single 2D image observation, and demonstrate that LFNs encode both scene geometry and appearance.

**Light fields and their reconstruction.** Light fields have a rich history as a scene representation in both computer vision and computer graphics. Adelson et al. [34] introduced the 5D plenoptic function as a unified representation of information in the early visual system [35]. Levoy et al. [36] and, concurrently, Gortler et al. [37] introduced light fields in computer graphics as a 4D sampled scene representation for fast image-based rendering. Light fields have since enjoyed popularity as a representation for novel view synthesis [38] and computational photography, e.g. [39]. Light fields enable direct rendering of novel views by simply extracting a 2D slice of the 4D light field. However, they tend to incur significant storage cost, and since they rely on two-plane parameterizations, they make it hard to achieve a full 360-degree representation without concatenating multiple light fields. A significant amount of prior work addresses reconstruction of fronto-parallel light fields via hand-crafted priors, such as sparsity in the Fourier or shearlet domains [40–42]. With the advent of deep learning, approaches to light field reconstruction that leverage convolutional neural networks to in-paint or extrapolate light fields from sparse views have been proposed [43, 7, 44], but similarly only support fronto-parallel novel view synthesis. We are instead interested in light fields as a representation of 3D appearance and geometry that enables efficient inference of and reasoning about the properties of the full underlying scene.

## 3 Background: 3D-structured Neural Scene Representations

Recent progress in neural scene representation and rendering has been driven by two key innovations. The first are neural fields, often also referred to as neural implicit- or coordinate-based scene representations $\Phi^{3D}$ [3, 4], which model a scene as a continuous function, parameterized as an MLP which maps a 3D coordinate to a representation $\mathbf{v}$ of whatever is at that 3D coordinate:

$$\Phi^{3D} : \mathbb{R}^3 \to \mathbb{R}^n, \quad \mathbf{x} \mapsto \Phi^{3D}(\mathbf{x}) = \mathbf{v}. \tag{1}$$

The second is a differentiable renderer $\mathbf{m}$, which, given a ray $\mathbf{r}$ in $\mathbb{R}^3$, and the representation $\Phi^{3D}$, computes the value of the color $\mathbf{c}$ of the scene when viewed along $\mathbf{r}$:

$$\mathbf{m}(\mathbf{r}, \Phi^{3D}) = \mathbf{c}(\mathbf{r}) \in \mathbb{R}^3. \tag{2}$$

Existing rendering methods broadly fall into two categories: sphere-tracing-based renderers [3, 45, 5, 46] and volumetric renderers [19, 4]. These methods require on the order of tens or hundreds of

evaluations of the values of $\Phi^{3D}$ along a ray $\mathbf{r}$ to compute $\mathbf{c}(\mathbf{r})$. This leads to extraordinarily large memory and time complexity of rendering. As training requires error backpropagation through the renderer, this impacts both training and test time.

# 4  The Light Field Network Scene Representation

We propose to represent a scene as a 360-degree *neural light field*, a function parameterized by an MLP $\Phi_\phi$ with parameters $\phi$ that directly maps the 4D space $\mathcal{L}$ of oriented rays to their observed radiance:

$$\Phi_\phi : \mathcal{L} \to \mathbb{R}^3, \mathbf{r} \mapsto \Phi_\phi(\mathbf{r}) = \mathbf{c}(\mathbf{r}). \tag{3}$$

A light field completely characterizes the flow of light through unobstructed space in a static scene with fixed illumination. Light fields have the unique property that rendering is achieved by a *single evaluation* of $\Phi$ per light ray, i.e., *no ray-casting is required*. Moreover, while the light field only encodes appearance explicitly, its derivatives encode geometry information about the underlying 3D scene [47, 34, 35]. This makes many methods to extract 3D geometry from light fields possible [48–51], and we demonstrate efficient recovery of sparse depth maps from LFNs below.

## 4.1  Implicit representations for 360 degree light fields

To fully represent a 3D scene requires a parameterization of all light rays in space. Conventional light field methods are constrained to leverage minimal parameterizations of the 4D space of rays, due to the high memory requirements of discretely sampled high-dimensional spaces. In contrast, our use of neural field representations allows us to freely choose a continuous parameterization that is mathematically convenient. In particular, we propose to leverage the 6D Plücker parameterization of the space of light rays $\mathcal{L}$ for LFNs. The Plücker coordinates (see [52] for an excellent overview) of a ray $\mathbf{r}$ through a point $\mathbf{p}$ in a normalized direction $\mathbf{d}$ are

$$\mathbf{r} = (\mathbf{d}, \mathbf{m}) \in \mathbb{R}^6 \text{ where } \mathbf{m} = \mathbf{p} \times \mathbf{d}, \text{ for } \quad \mathbf{d} \in \mathbb{S}^2, \mathbf{p} \in \mathbb{R}^3. \tag{4}$$

where $\times$ denotes the cross product. While Plücker coordinates are a-priori 6-tuples of real numbers, the coordinates of any ray lie on a curved 4-dimensional subspace $\mathcal{L}$. Plücker coordinates uniformly represent all oriented rays in space without singular directions or special cases. Intuitively, a general ray $\mathbf{r}$ together with the origin define a plane, and $\mathbf{m}$ is a normal vector to the plane with its magnitude capturing the distance from the ray to the origin; if $\mathbf{m} = 0$ then the ray passes through the origin and is defined by its direction $\mathbf{d}$. This is in contrast to conventional light field parameterizations: Fronto-parallel two-plane or cylindrical parameterizations cannot represent the full 360-degree light field of a scene [36, 53]. Cubical two-plane arrangements [37, 38] are not continuous, complicating the parameterization via a neural implicit representation. In contrast to the two-sphere parameterization [54], Plücker coordinates do not require that scenes are bounded in size and do not require spherical trigonometry.

The parameterization via a neural field enables compact storage of a 4D light field that can be sampled at arbitrary resolutions, while non-neural representations are resolution-limited. Neural fields further allow the analytical computation of derivatives. This enables the efficient computation of sparse depth maps, where prior representations of light fields require finite-differences approximations of the gradient [48–50].

**Rendering LFNs.**  To render an image given an LFN, one computes the Plücker coordinates $\mathbf{r}_{u,v}$ of the camera rays at each $u, v$ pixel coordinate in the image according to Equation 4. Specifically, given the extrinsic $\mathbf{E} = \begin{bmatrix} \mathbf{R}|\mathbf{t} \end{bmatrix} \in SE(3)$ and intrinsic $\mathbf{K} \in \mathbb{R}^{3\times3}$ camera matrices [55] of a camera, one may retrieve the Plücker coordinates of the ray $\mathbf{r}_{u,v}$ at pixel coordinate $u, v$ as:

$$\mathbf{r}_{u,v} = (\mathbf{d}_{u,v}, \mathbf{t} \times \mathbf{d}_{u,v}) / \|\mathbf{d}_{u,v}\|, \text{ where } \mathbf{d}_{u,v} = \mathbf{R}\mathbf{K}^{-1} \begin{pmatrix} u \\ v \\ 1 \end{pmatrix} + \mathbf{t}, \tag{5}$$

where we use the world-to-camera convention for the extrinsic camera parameters. Rendering then amounts to a *single* evaluation of the LFN $\Phi$ for each ray, $\mathbf{c}_{u,v} = \Phi(\mathbf{r}_{u,v})$. For notational convenience, we introduce a rendering function

$$\Theta_{\mathbf{E},\mathbf{K}}^{\Phi} : \mathbb{R}^\ell \to \mathbb{R}^{H \times W \times 3} \tag{6}$$

which renders an LFN $\Phi_\phi$ with parameters $\phi \in \mathbb{R}^\ell$ when viewed from a camera with extrinsic and intrinsic parameters $(\mathbf{E}, \mathbf{K})$ into an image.

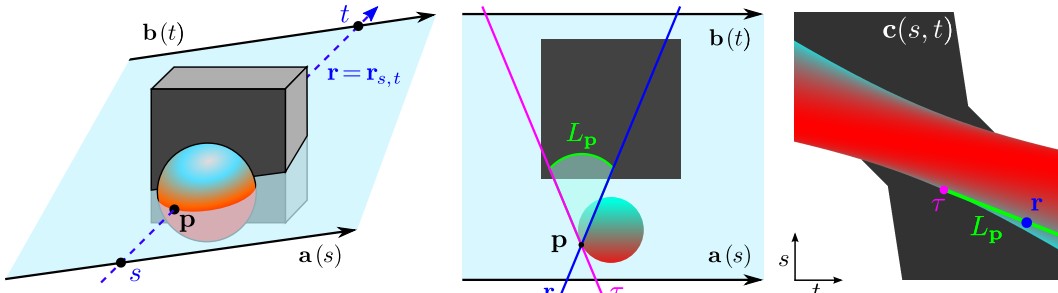

Figure 2: Given a 3D scene (left) and a light ray $\mathbf{r}$ (blue), we can slice the scene along a 2D plane containing the ray (light blue), yielding a 2D scene (center). The light field of all rays in a 2D plane, the *Epipolar Plane Image* (EPI) $\mathbf{c}(s, t)$ (right). can be analytically computed from our 360-degree LFN. The family of rays $L_{\mathbf{p}}$ (green) going through a point $\mathbf{p}$ on an object in the scene defines a straight line in the EPI. See below for further discussion of EPI geometry.

## 4.2 The geometry of Light Field Networks

We will now analyze the properties of LFNs representing Lambertian 3D scenes, and illustrate how the geometry of the underlying 3D scene is encoded. We will first derive an expression that establishes a relationship between LFNs and the classic two-plane parameterization of the light field. Subsequently, we will derive an expression for the depth of a ray in terms of the local color gradient of the light field, therefore allowing us to efficiently extract sparse depth maps from the light field at any camera pose via analytical differentiation of the neural implicit representation. Please see Figure 2 for an overview.

**Locally linear slices of the light field.** We derive here a local parametrization that will allow us to work with an LFN as if it were a conventional 2-plane light field. Given a ray $\mathbf{r}$ in Plücker coordinates, we pick two points $\mathbf{x}, \mathbf{x}' \in \mathbb{R}^3$ along this ray. We then find a normalized direction $\mathbf{d} \in \mathbb{S}^2$ not parallel to the ray direction - a canonical choice is a direction orthogonal to the ray direction. We may now parameterize two parallel lines $\mathbf{a}(s) = \mathbf{x} + s\mathbf{d}$ and $\mathbf{b}(t) = \mathbf{x}' + t\mathbf{d}$ that give rise to a local two-plane basis of the light field with ray coordinates $s$ and $t$. $\mathbf{r}$ intersects these lines at the two-plane coordinates $(s, t) = (0, 0)$. This choice of local basis now assigns the two-plane coordinates $(s, t)$ to the ray $\mathbf{r}$ from $\mathbf{a}(s)$ to $\mathbf{b}(t)$. In Figure 2, we illustrate this process on a simple 2D scene.

**Epipolar Plane Images and their geometry.** The Plücker coordinates (see Eq. 4) enable us to extract a 2D slice from an LFN field by varying $(s, t)$ and sampling $\Phi$ on the Plücker coordinates of the rays parametrized pairs of points on the lines $\mathbf{a}(s)$ and $\mathbf{b}(t)$:

$$\mathbf{c}(s, t) = \Phi\left(\mathbf{r}(s, t)\right), \text{ where } \mathbf{r}(s, t) = \overrightarrow{\mathbf{a}(s)\mathbf{b}(t)} = \left( \frac{\mathbf{b}(t) - \mathbf{a}(s)}{\|\mathbf{b}(t) - \mathbf{a}(s)\|}, \frac{\mathbf{a}(s) \times \mathbf{b}(t)}{\|\mathbf{b}(t) - \mathbf{a}(s)\|} \right). \quad (7)$$

The image of this 2D slice $\mathbf{c}(s, t)$ is well-known in the light field literature as an *Epipolar Plane Image* (EPI) [47]. EPIs carry rich information about the geometry of the underlying 3D scene. For example, consider a point $\mathbf{p}$ on the surface of an object in the scene; please see Figure 2 for a diagram. A point $\mathbf{p} \in \mathbb{R}^2$ has a 1-dimensional family of rays going through the point, which correspond to a (green) line $L_{\mathbf{p}}$ in the EPI. In a Lambertian scene, all rays that meet in this point and that are not occluded by other objects must observe the same color. Therefore, the light field is constant along this line. As one travels along $L_{\mathbf{p}}$, rotating through the family of rays through $\mathbf{p}$, one eventually reaches a (magenta) *tangent ray* $\tau$ to the object. At a tangent ray, the value of the EPI ceases to be constant, and the light field changes its color to whatever is disoccluded by the object at this tangent ray. Because objects of different depth undergo differing amounts of parallax, the *slope* of the segment of $L_{\mathbf{p}}$ along which $\mathbf{c}$ is constant determines the 3D coordinates of $\mathbf{p}$. Finally, by observing that we may extract EPIs from *any* perspective, it is clear that an LFN encodes the full 3D geometry of the underlying scene. Intuitively, this may also be seen by con-

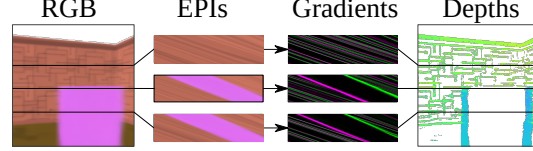

sidering that one could render out all possible perspectives of the underlying scene, and solve a classic multi-view stereo problem to retrieve the shape.

**Extracting depth maps from LFNs.** A correctly inferred light field necessarily contains accurate 3D geometry information, although the geometry is encoded in a nontrivial way. To extract 3D geometry from an LFN, we utilize the property of the 2-plane parameterization that the light field is constant on segments $L_\mathbf{p}$, the slopes of which determine $\mathbf{p}$. In the supplemental material, we derive

**Proposition 1.** For a Lambertian scene, the distance $d$ along $\mathbf{r} = \overrightarrow{\mathbf{a}(s)\mathbf{b}(t)}$ from $\mathbf{a}(s)$ to the point $\mathbf{p}$ on the object is

$$d(\mathbf{r}) = D \frac{\partial_t \mathbf{c}(s,t)}{\partial_s \mathbf{c}(s,t) + \partial_t \mathbf{c}(s,t)}. \tag{8}$$

where $\mathbf{a}(s)$ and $\mathbf{b}(t)$ are as above, $\mathbf{c}(s,t)$ is defined by (7), $D$ is the distance between the lines $\mathbf{a}(t)$ and $\mathbf{b}(t)$. Thus $\mathbf{p} = \mathbf{a}(s) + d(\mathbf{r}) \frac{\mathbf{b}(t) - \mathbf{a}(s)}{\|\mathbf{b}(t) - \mathbf{a}(s)\|}$, and $\partial_x$ denotes the partial derivative by variable $x$.

This result yields meaningful depth estimates wherever the derivatives of the light fields are nonzero along the ray. In practice, we sample several rays in a small $(s,t)$ neighborhood of the ray $\mathbf{r}$ and declare depth estimates as invalid if the gradients have high variance-please see the code for implementation details. This occurs when $\mathbf{r}$ hits the object at a point where the surface color is changing, or when $\mathbf{r}$ is a tangent ray. We note that there is a wealth of prior art that could be used to extend this approach to extract *dense* depth maps [48–51].

## 4.3 Meta-learning with conditional Light Field Networks

We consider a dataset $\mathcal{D}$ consisting of $N$ 3D scenes

$$S_i = \{(\mathbf{I}_j, \mathbf{E}_j, \mathbf{K}_j)\}_{j=1}^K \in \mathbb{R}^{H \times W \times 3} \times SE(3) \times \mathbb{R}^{3 \times 3}, \; i = 1 \ldots N \tag{9}$$

with $K$ images $\mathbf{I}_j$ of each scene taken with cameras with extrinsic parameters $\mathbf{E}_j$ and intrinsic parameters $\mathbf{K}_j$ [55]. Each scene is completely described by the parameters $\phi_i \in \mathbb{R}^\ell$ of its corresponding light field MLP $\Phi_i = \Phi_{\phi_i}$.

**Meta-learning and multi-view consistency.** In the case of 3D-structured neural scene representations, ray-marching or volumetric rendering naturally ensure multi-view consistency of the reconstructed 3D scene representation. In contrast, a general 4D function $\Phi : \mathcal{L} \to \mathbb{R}^3$ is not multi-view consistent, as most such functions are not the light fields of any 3D scene. We propose to overcome this challenge by learning a prior over the space of light fields. As we will demonstrate, this prior can also be used to reconstruct an LFN from a single 2D image observation. In this paradigm, differentiable ray-casting is a method to force the light field of a scene to be multi-view consistent, while *we instead impose multi-view consistency by learning a prior over light fields*.

**Meta-learning framework.** We propose to represent each 3D scene $S_i$ by its own latent vector $\mathbf{z}_i \in \mathbb{R}^k$. Generalizing to new scenes amounts to learning a prior over the space of light fields that is concentrated on the manifold of multi-view consistent light fields of natural scenes. To represent this latent manifold, we utilize a hypernetwork [56, 3]. The hypernetwork is a function, represented as an MLP

$$\Psi : \mathbb{R}^k \to \mathbb{R}^\ell, \Psi_\psi(\mathbf{z}_i) = \phi_i \tag{10}$$

with parameters $\psi$ which sends the latent code $\mathbf{z}_i$ of the $i$-th scene to the parameters of the corresponding LFN.

Several reasonable approaches exist to obtain latent codes $\mathbf{z}_i$. One may leverage a convolutional- or transformer-based image encoder, directly inferring the latent from an image [11, 5], or utilize gradient-based meta-learning [23]. Here, we follow an auto-decoder framework [1, 3] to find the latent codes $\mathbf{z}_i$, but note that LFNs are in no way constrained to this approach. We do not claim that this particular meta-learning method will out-perform other forms of conditioning, such as gradient-based meta-learning [57, 23] or FILM conditioning [58], but perform a comparison to a conditioning-by-concatenation approach in the appendix. We assume that the latent vectors have

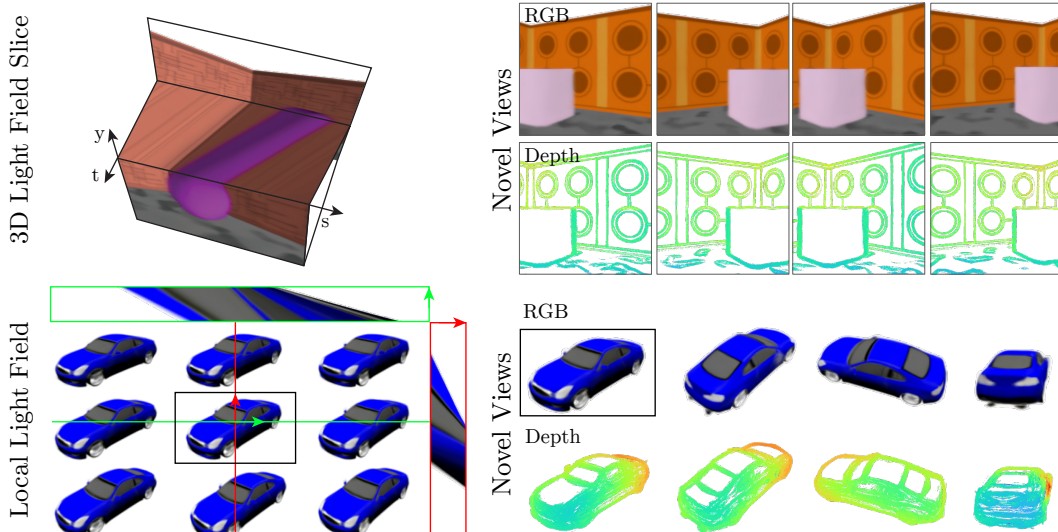

Figure 3: **360-degree light field parameterization.** Top left: 3D slice in the style of the Lumi-graph [37] of an LFN of a room-scale scene. Bottom left: nearby views of a car as well as vertical (right, red) and horizontal (top, green) Epipolar Plane Images, sampled from an LFN. Reconstructed from training set, 50 and 15 views, respectively. Right: LFNs enable rendering from arbitrary, 360-degree camera perspectives as well as sparse depth map extraction from only a single sample per ray. Please see the supplemental video for more qualitative results.

a Gaussian prior with zero mean and a diagonal covariance matrix. At training time, we jointly optimize the latent parameters $\mathbf{z}_i$ together with the hypernetwork parameters $\psi$ using the objective

$$\arg\min_{\{\mathbf{z}_i\},\psi} \sum_i \sum_j \|\Theta^{\Phi}_{\mathbf{E}_j,\mathbf{K}_j}(\Psi_\psi(\mathbf{z}_i)) - \mathbf{I}_j\|_2^2 + \lambda_{lat}\|\mathbf{z}_i\|_2^2. \tag{11}$$

Here the $\Theta^\Phi$ is the rendering function (Equation 6), the first term is an $\ell_2$ loss penalizing the light fields that disagree with the observed images, and the second term enforces the prior over the latent variables. We solve Equation 11 using gradient descent. At test time, we freeze the parameters of the hypernetwork and reconstruct the light field for a new scene $S$ given a *single observation* of the scene $\{(\mathbf{I}, \mathbf{E}, \mathbf{K})\}$ by optimizing, using gradient descent, the latent variable $\mathbf{z}_S$ of the scene, such that the reconstructed light field $\Phi_{\Psi_\psi(\mathbf{z}_S)}$ best matches the given observation of the scene:

$$\mathbf{z}_S = \arg\min_{\mathbf{z}} \|\Theta^{\Phi}_{\mathbf{E},\mathbf{K}}(\Psi_\psi(\mathbf{z})) - \mathbf{I})\|_2^2 + \lambda_{lat}\|\mathbf{z}\|_2^2. \tag{12}$$

**Global vs. local conditioning**  The proposed meta-learning framework globally conditions an LFN on a single latent variable $\mathbf{z}$. Recent work instead leverages *local* conditioning, where a neural field is conditioned on local features extracted from a context image [26, 6, 27]. In particular, the recently proposed pixelNeRF [6] has achieved impressive results on few-shot novel view synthesis. As we will see, the current formulation of LFNs does *not* outperform pixelNeRF. We note, however, that local conditioning methods solve a *different problem*. Rather than learning a prior over classes of *objects*, local conditioning methods learn priors over *patches*, answering the question "How does this image patch look like from a different perspective?". As a result, this approach does not learn a latent space of neural scene representations. Rather, scene context is required to be available at test time to reason about the underlying 3D scene, and the representation is not *compact*: the size of the conditioning grows with the number of context observations. In contrast, globally conditioned methods [3, 11, 1, 2] first infer a *global* representation that is invariant to the number of context views and subsequently discard the observations. However, local conditioning enables better generalization due to the shift-equivariance of convolutional neural networks. An equivalent to local conditioning in light fields is non-obvious, and an exciting direction for future work.

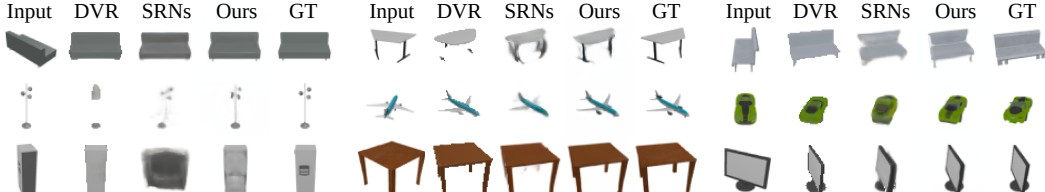

| Input | DVR | SRNs | Ours | GT | Input | DVR | SRNs | Ours | GT | Input | DVR | SRNs | Ours | GT |

Figure 4: **Category-agnostic single-shot reconstruction.** We train a single model on the 13 largest Shapenet classes. LFNs consistently outperform global conditioning baselines. Baseline results courtesy of the pixelNeRF [6] authors. Please see the supplemental video for more qualitative results.

Table 1: **Single-shot multi-class reconstruction results.** We benchmark LFNs with SRNs [3] and DVR [5] on the task of single-shot (auto-decoding with a single view), multi-class reconstruction of the 13 largest ShapeNet [59] classes. LFNs significantly outperform DVR and SRNs on almost all classes, on average by more than 1dB, while only requiring a *single* network evaluation per ray. Note that DVR requires additional supervision in the form of foreground-background masks. Means across classes are weighted equally.

| | | plane | bench | cbnt. | car | chair | disp. | lamp | spkr. | rifle | sofa | table | phone | boat | mean |
|---|---|---|---|---|---|---|---|---|---|---|---|---|---|---|---|
| ↑ PSNR | DVR [5] | 25.29 | 22.64 | 24.47 | 23.95 | 19.91 | **20.86** | 23.27 | 20.78 | 23.44 | 23.35 | 21.53 | **24.18** | 25.09 | 22.70 |
| | SRN [3] | 26.62 | 22.20 | 23.42 | 24.40 | 21.85 | 19.07 | 22.17 | 21.04 | 24.95 | 23.65 | 22.45 | 20.87 | 25.86 | 23.28 |
| | LFN | **29.95** | **23.21** | **25.91** | **28.04** | **22.94** | 20.64 | **24.56** | **22.54** | **27.50** | **25.15** | **24.58** | 22.21 | **27.16** | **24.95** |
| ↑ SSIM | DVR [5] | 0.905 | **0.866** | **0.877** | 0.909 | 0.787 | **0.814** | **0.849** | 0.798 | 0.916 | 0.868 | 0.840 | **0.892** | 0.902 | 0.860 |
| | SRN [3] | 0.901 | 0.837 | 0.831 | 0.897 | 0.814 | 0.744 | 0.801 | 0.779 | 0.913 | 0.851 | 0.828 | 0.811 | 0.898 | 0.849 |
| | LFN | **0.932** | 0.855 | 0.871 | **0.943** | **0.835** | 0.786 | 0.844 | **0.808** | **0.939** | **0.874** | **0.868** | 0.844 | **0.907** | **0.870** |

# 5   Experiments

We demonstrate the efficacy of LFNs by reconstructing 360-degree light fields of a variety of simple 3D scenes. In all experiments, we parameterize LFNs via a 6-layer ReLU MLP, and the hypernetwork as a 3-layer ReLU MLP, both with layer normalization. We solve all optimization problems using the ADAM solver with a step size of $10^{-4}$. Please find more results, as well as precise hyperparameter, implementation, and dataset details, in the supplemental document and video.

**Reconstructing appearance and geometry of single-object and room-scale light fields.**   We demonstrate that LFN can parameterize 360-degree light fields of both single-object ShapeNet [59] objects and simple, room-scale environments. We train LFNs on the ShapeNet "cars" dataset with 50 observations per object from [3], as well as on simple room-scale environments as proposed in [13]. Subsequently, we evaluate the ability of LFNs to generate novel views of the underlying 3D scenes. Please see Figure 3 for qualitative results. LFNs succeed in parameterizing the 360-degree light field, enabling novel view synthesis at *real-time* frame-rates (see supplemental video). We further demonstrate that LFNs encode scene geometry by presenting Epipolar Plane Images and leveraging the relationship derived in Equation 8 to infer sparse depth maps. We highlight that both rendering and depth map extraction *do not require ray-casting*, with only a single evaluation of the network or the network and its gradient respectively.

**Multi-class single-view reconstruction.**   Following [5, 6], we benchmark LFNs with recent *global conditioning methods* on the task of single-view reconstruction and novel view synthesis of the 13 largest ShapeNet categories. We follow the same evaluation protocol as [60] and train a single model across all categories. See Figure 4 for qualitative and Table 1 for quantitative baseline comparisons. We significantly outperform both Differentiable Volumetric Rendering (DVR) [5] and Scene Representation Networks (SRNs) [3] on all but two classes by an average of 1dB, while requiring more than an order of magnitude fewer network evaluations per ray. Qualitatively, we find that the reconstructions from LFNs are often crisper than those of either Scene Representation Networks or DVR. Note that DVR requires additional ground-truth foreground-background segmentation masks.

**Class-specific single-view reconstruction.**   We benchmark LFNs on single-shot reconstruction on the Shapenet "cars" and "chairs" classes as proposed in SRNs [3]. See Figure 5 for qualitative and quantitative results. We report performance better than SRNs in PSRN and on par in terms of SSIM

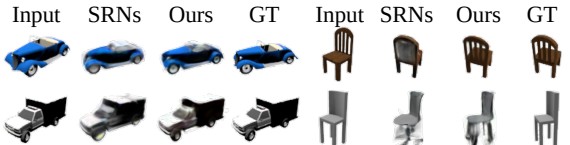

| | Input | SRNs | Ours | GT | Input | SRNs | Ours | GT |
|---|---|---|---|---|---|---|---|---|

| | Chairs | Cars |
|---|---|---|
| SRNs [3] | **22.89** / 0.89 | 22.25 / **0.89** |
| LFN | 22.26 / **0.90** | **22.42** / 0.89 |

Figure 5: **Class-specific single-shot reconstruction.** LFN performs approximately on par with SRNs [3] in the single-class single-shot reconstruction case, while requiring an order of magnitude fewer network evaluations, memory, and rendering time. Quantitative results report (PSNR, SSIM).

| | Class-specific | Multi-Class |
|---|---|---|
| LFN | 22.34 / 0.90 | 24.95 / 0.87 |
| pixelNeRF | **23.45** / **0.91** | **26.80** / **0.91** |

Figure 6: **Global vs. local conditioning.** The locally conditioned pixelNeRF outperforms LFNs in single-shot reconstruction, though LFNs require *three orders of magnitude less rendering time and memory*. Qualitatively, for many objects LFNs are on par with pixelNERF (left), but confuse the object class on others (right). Quantitative results report (PSNR, SSIM). 'Class-specific' refers to average over cars and chairs as in Fig. 5 while 'multi-class' refers to the average over all ShapeNet classes as in Fig. 1.

on the "cars" class, and worse in PSNR but better in terms of SSIM on the "chairs" class, while requiring an order of magnitude fewer network evaluations and rendering in real-time. We attribute the drop in performance compared to multi-class reconstruction to the smaller dataset size, causing multi-view inconsistency.

**Global vs. local conditioning and comparison to pixelNeRF [6].** We investigate the role of global conditioning, where a single latent is inferred to describe the whole scene [3], to local conditioning, where latents are inferred *per-pixel* in a 2D image and leveraged to locally condition a neural implicit representation [26, 27, 6]. We benchmark with the recently proposed pixelNeRF [6]. As noted above (see Section 4.3), local conditioning does *not* infer a compact neural scene representation of the scene. Nevertheless, we provide the comparison here for completeness. See Figure 6 for qualitative and quantitative results. On average, LFNs perform 1dB worse than pixelNeRF in the single-class case, and 2dB worse in the multi-class setting.

**Real-time rendering and storage cost.** See Table 2 for a quantitative comparison of the rendering complexity of LFN compared with that of volumetric and ray-marching based neural renderers [3, 45, 19, 4, 6]. All clock times were collected for rendering $256 \times 256$ images on an NVIDIA RTX 6000 GPU. We further compare the cost of storing a single LFN with the cost of storing a conventional light field. With approximately 400k parameters, a single LFN requires around 1.6 MB of storage, compared to 146 MB required for storing a 360-degree light field at a resolution of $256 \times 256 \times 17 \times 17$ in the six-plane Lumigraph configuration.

**Multi-view consistency as a function of training set size.** We investigate how multi-view consistency scales with the amount of data that the prior is trained on. Please find this analysis in the supplementary material.

Table 2: **Comparison of rendering complexity.** LFNs require *three orders of magnitude* less compute than volumetric rendering based approaches, and admit real-time rendering. Please see Table of supplemental material for comparison on larger images based on data in []

| | LFNs | SRNs [3] | pixelNeRF [6] |
|---|---|---|---|
| # evaluations per ray | 1 | 11 | 192 |
| clock time for $256 \times 256$ image (ms) | 2.1 | 120 | $30e3$ |

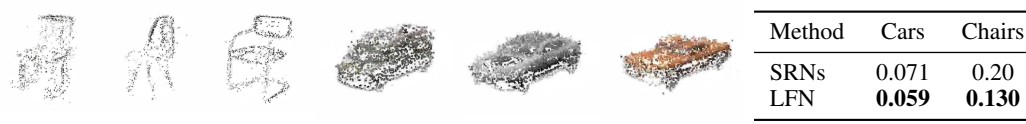

| Method | Cars | Chairs |
|--------|------|--------|
| SRNs | 0.071 | 0.20 |
| LFN | **0.059** | **0.130** |

Figure 7: **Geometry reconstruction with LFNs.** By backprojecting extracted depth into 3D, LFNs enable reconstruction of 3D pointclouds (left). For rays were depth is valid, LFNs achieve lower mean L1 depth error than Scene Representation Networks (right).

**Overfitting of single 3D scenes.**    We investigate overfitting a single 3D scene with a Light Field Network with positional encodings / sinusoidal activations [24, 61]. Please find this analysis in the supplementary material.

**Evaluation of Reconstructed Geometry.**    We investigate the quality of the geometry that can be computed from an LFN via Eq. 8. For every sample in the class-specific single-shot reconstruction experiment, we extract its per-view sparse depth map. We then backproject depth maps from four views into 3D to reconstruct a point cloud, and benchmark mean L1 error on valid depth estimates with Scene Representation Networks [3]. Fig. 7 displays qualitative and quantitative results. Qualitatively, point clouds succeed in capturing fine detail such as the armrests of chairs. Quantitatively, LFNs outperform SRNs on both cars and chairs. We note that LFNs have a slight advantage in this comparison, as we can only benchmark on the sparse depth values, for which LFNs have high confidence. This includes occlusion boundaries, which are areas where the sphere-tracing based SRNs incurs high error, as it is forced to take smaller and smaller steps and may not reach the surface. We highlight that we do not claim that the proposed method is competetive with methods designed for geometry reconstruction in particular, but that we only report this to demonstrate that the proposed method is capable to extract valid depth estimates from an LFN.

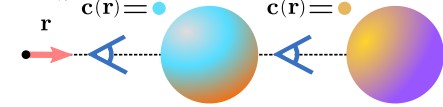

**Limitations.**    First, as every existing light field approach, LFNs store only one color per oriented ray, which makes rendering views from cameras placed in between occluding objects challenging, even if the information may still be stored in the light field. Second, though we outperform globally-conditioned methods, we currently do not outperform the locally-conditioned pixelNeRF. Finally, as opposed to 3D-structured representations, LFNs do not enforce strict multi-view consistency, and may be inconsistent in the case of small datasets.

## 6    Discussion and Conclusion

We have proposed Light Field Networks, a novel neural scene representation that directly parameterizes the full 360-degree, 4D light field of a 3D scene via a neural implicit representation. This enables both real-time neural rendering with a *single* evaluation of the neural scene representation per ray, as well as sparse depth map extraction without ray-casting. Light Field Networks outperform globally conditioned baselines in single-shot novel view synthesis, while being three orders of magnitude faster and less memory-intensive than current volumetric rendering approaches. Exciting avenues for future work include combining LFNs with local conditioning, which would enable stronger out-of-distribution generalization, studying the learning of non-Lambertian scenes, and enabling camera placement in obstructed 3D space. With this work, we make important contributions to the emerging fields of neural rendering and neural scene representations, with exciting applications across computer vision, computer graphics, and robotics.

**Societal Impacts.**    Potential improvements extending our work on few-observation novel view synthesis could enable abuse by decreasing the cost of non-consensual impersonations. We refer the reader to a recent review of neural rendering [22] for an in-depth discussion of this topic.

## Acknowledgements and Disclosure of Funding

This work is supported by the NSF under Cooperative Agreement PHY-2019786 (The NSF AI Institute for Artificial Intelligence and Fundamental Interactions, http://iaifi.org/), ONR under 1015

G TA243/N00014-16-1-2007 (Understanding Scenes and Events through Joint Parsing, Cognitive Reasoning and Lifelong Learning), Mitsubishi under 026455-00001 (Building World Models from some data through analysis by synthesis), DARPA under CW3031624 (Transfer, Augmentation and Automatic Learning with Less Labels), as well as the Singapore DSTA under DST00OECI20300823 (New Representations for Vision). We thank Andrea Tagliasacchi, Tomasz Malisiewicz, Prafull Sharma, Ludwig Schubert, Kevin Smith, Bernhard Egger, Christian Richardt, Manuel Rey Area, and Jürgen and Susanne Sitzmann for interesting discussions and feedback, and Alex Yu for kindly sharing the outputs of pixelNeRF and baselines with us.

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
