# OpenReview forum: "Light Field Networks: Neural Scene Representations with Single-Evaluation Rendering"
_NeurIPS.cc/2021/Conference — NeurIPS 2021 Spotlight_

### Official Review · Reviewer_NEE3 · 2021-06-28

**Rating:** 8
**Confidence:** 4

**Summary:**

Light Field Networks encode the light field of a 3D scene, which comes with certain restrictions on where an observer can be placed. LFNs are coordinate-based MLPs that use the Plucker parameterization of directed rays in 3D space to represent rays. Thus they can output the final color of a pixel with a single network evaluation, massively speeding up rendering relative to NeRF but sacrificing "hard" multi-view consistency, which now needs to be learned. LFNs here are trained via a hypernetwork and scene conditioning happens via an auto-decoded latent code for the hypernetwork. While light fields only represent appearance directly, it is possible to use gradients to extract depth along appearance edges. Results are presented on ShapeNet and very simple synthetic rooms.

**Limitations And Societal Impact:**

Limitations are addressed well and potential negative societal impact is shortly mentioned.

**Main Review:**

- LFNs are novel and an interesting take on neural radiance fields that trades multi-view-consistency-by-design for orders of magnitude higher rendering speed.
- Results in the supplemental video are random and not cherry-picked.
- Qualitative results seem overall similar to DVR to me. But since LFNs are much faster, they offer an advantage over DVR in terms of results.
- The paper is well written.

- I would have liked a geometrical intuition as to why Plucker coordinates are independent of the specific point on the ray. The origin and the ray (i.e. all points p on the ray) define one single plane and p x d is the normal vector of that plane. Rays through the origin are a corner case in that intuition, but they are directly identified by their direction.

- Interestingly, when comparing single-class vs. multi-class, chairs have better results overall on single-class while cars are better in multi-class. Looking at more classes in single-class experiments would give some indication on whether the claim in line 280 (smaller dataset is the issue) is correct, which would mean that chairs is an outlier class.
- How many objects per class are there roughly in the single-class and multi-class settings? Hundreds?
- It would also be interesting to see how multi-view consistency (e.g. measured naively via PSNR/SSIM of novel views on a random but fixed set of e.g. 10 shapes per class) improves with an increasing training set size (e.g. 1, 10, 100, 1k, all per class), both in the single-class and multi-class setting. That would give a better idea on how much the hypernetwork learns multi-view consistency as an abstract property, which is the main motivation for using a hypernetwork (lines 10-12).
- In general, more experiments than just straightforward reconstruction results would add insights. E.g. an ablation of the hypernetwork setting vs. a simpler latent-code conditioning setting with a single, larger LFN. Or a quantitative evaluation of the sparse depth maps, even if only at the sparse points where depth is extractable. Especially the latter should be added since all of page 5 and Sec. 1 of the supplement discuss depth extraction. A comparison to DVR and SRNs, if possible, would offer a baseline. Or backproject multiple depth renderings of a single object and show the merged point cloud in the video, for example. Some experiment that gives a better idea of how well geometry extraction works.
- Since light fields mostly support inside-out scenes similar to the room scenes and outside-in scenes like the ShapeNet objects, and since both settings are spherical, a simple two-sphere parametrization (e.g. a view-conditionined NeRF that is only evaluated once on a single 3D sphere) would be enough for the results presented in the paper. Better scenes that highlight the strengths of the particular parametrization would have been nice, namely unbounded scenes, e.g. some very long corridor (lines 141-142).

Overall:

LNFs take a valuable route towards better neural graphics representations that is complementary to the slow and hard-constrained setting of NeRF. I hope that future work will be able to re-introduce the multi-view consistency by design instead of learning it as a soft constraint. The main issue I have with the paper in its current form is the experimental investigation. I believe that depth evaluation is necessary, that an experiment investigating multi-view consistency vs. training set size is at least strongly recommendable, and that a hypernetwork vs. simple conditioning experiment and evaluating scenes benefiting from the Plucker parametrization would strengthen the paper noticeably but aren't required for acceptance.

Major clarity:

- What are class-specific and multi-class in the table in Fig. 6 referring to? Means across all classes? Or what class was evaluated for the class-specific setting?
- Related to that, means across all classes (weighted equally or by class size, should be mentioned) could be added to Table 1.
- line 299: "more difficult" sounds too inaccurate to me. It is only even possible in a few carefully designed cases (e.g. line 68 in the supplement) and otherwise simply impossible (e.g. placing an arbitrarily rotating observer inside the convex hull of an outside-in scene like chairs from ShapeNet).
- For single-class single-shot car reconstruction, the video states that LFNs offer slightly more detail, but I am unable to confirm that. To my eye, LFNs and SRNs are qualitatively on par and quantitatively LFNs are only slightly better. That is okay considering the speed difference.
- How long does auto-decoding in the single-view setting take per scene at test time? Lines 86 and 97 in the supplement state "until convergence". How long does that take in practice?

Minor clarity:

- "Moreoever" in line 123
- I don't understand why line 212 is emphasized. Wouldn't line 213 be more appropriate?
- Eq. 10: there's an unnecessary closing paranthesis (same in Eq. 11) and it should be z_i in the second term. It would also be easier to parse if the arg min is over \{z_i\} instead of z_i.
- What does "single-shot" in Table 1 refer to? Auto-decoding with a single view?
- Table 1 boat SSIM is incorrectly bolded.
- Fig. 5 Cars SSIM: LFN and SRNs are on par w.r.t. the two reported digits. That's wrongly bolded and incorrectly stated in line 277.

----------

The authors have provided a very thorough rebuttal. They addressed my concerns very well. I read the other reviewers' concerns and I am satisfied by the authors' response to those. Given the improvements from the rebuttal, I have increased my score from a 5 to an 8.

This work opens up a number of interesting directions, I especially liked the remark at the end of Sec. 1.4 of the rebuttal: "We argue that LFNs offer an intriguing orthogonal approach to neural rendering that may in future work inform or even be combined with 3D-structured neural representations."

The only concern that remains for me with the work is Sec. 2.5 in the rebuttal regarding real-world scenes, which I did not find as convincing as the other points. That LFNs struggle in the overfitting regime, which general real-world scenes fall into, is an unfortunate downside that future work hopefully addresses.

**Time Spent Reviewing:**

4

---

> ### Author Response · Authors · 2021-08-10
> **Response to NEE3**
>
> We thank reviewer NEE3 for their careful analysis and specific, actionable feedback! We agree that the proposed experiments will add to the paper significantly, and are happy that we were able to address almost all of the reviewer’s requested experiments. We respond to the comments, in order - **we regularly refer to the response overview, which is the top-level post in this forum**:
>
> >a geometrical intuition as to why Plucker coordinates are independent of the specific point on the ray
>
>
> This is a nice way of putting it, and we will add this geometrical description after line 135, following the introduction of Plucker coordinates. Thanks!
>
> >How many objects per class are there roughly in the single-class and multi-class settings? Hundreds?
>
>
> In the single-class setting, there are 4.5k cars and 2.6k chairs. In the multi-class setting, there are approximately 2.3k objects per class, on average.
>
> **Multi-view consistency vs. training set size.** We strongly agree that these are very interesting questions, and we are happy to report that we were able to evaluate multi-view consistency vs training set size exactly as the reviewer recommends, as detailed in section 2.4 of the response overview. We found that multi-view consistency measured as the reviewer suggested increases monotonically with training-set size. To disentangle the effects of learning a better prior over the given objects as opposed to learning a better generic multi-view consistency prior, we ran a further experiment, detailed in the second half of section 2.4, which shows similar monotonic improvements of the multi-view consistency prior with training set size. We hope that these results are of interest, and we will add them into the paper!
>
> **Ablation of the hypernetwork setting vs. a simpler latent-code conditioning setting.**
> This was requested by other reviewers as well, and we detail the results of our ablation experiment comparing with conditioning via concatenation in section 2.3 of our response overview. We found that hypernetworks have superior performance; however, we do not claim that our conditioning method outperforms other conditioning methods such as FILM-conditioning, and we think that a systematic evaluation of LFNs with varying conditioning methods would be a worthwhile project. Please see section 2.3 of the overview for more discussion.
> >unbounded scenes, e.g. some very long corridor.
>
> We agree that this is a great way to demonstrate the effectiveness of the Plucker parameterization of 360-degree light fields, and we are currently running an experiment on a dataset of corridors of varying length, size, wall and floor texture, built from the same building blocks as the GQN rooms. We will add the results of this experiment to the final version of the paper. Thank you for the suggestion!
>
> **Depth evaluation:**
> We fully agree that quantitative evaluation of the quality sparse depth maps will make the paper stronger, and are happy to report that we **performed both qualitative and quantitative evaluation of geometry extraction**.
>
> For quantitative evaluation, we benchmark mean L1 error on valid depth estimates of SRNs with depths from SRNs, which has been confirmed as a strong baseline for depth estimation on Facebook AI’s newly released CO3D dataset [1]. Please see 2.1 of the response overview for the results! __We outperform SRNs, on both cars and the chairs datasets__. However, we note that the purpose of this evaluation is only to establish empirically that depth information is indeed *encoded* in LFNs, and that we see the development of superior depth extraction methods as an exciting avenue for future work. We thank the reviewer for suggesting that we perform this evaluation, and will add it into the paper.
>
> For qualitative depth evaluation, we will attach videos to the supplemental material in which we back-project the sparse depth maps into 3D and color the resulting point cloud according to the RGB color at each point. Please see Section 2.2 of the response overview for preliminary results and details of the material to be released!
>
> We now address the major and minor points of clarification:
>
> **Major clarifications:**
> Single vs. Multi-Class: for space reasons, we took the average over PSNR and SSIM values over cars and chairs for the Single-Class and the 13 shapenet classes in the multi-class setting.
> Means across all classes are weighted equally, we will add this to Table 1.
> Placing camera between occluders is impossible: We agree that using the current formulation of LFNs, this is impossible, and we will clarify this in the paper. The formulation was meant to highlight that this information *is* stored in the light field (see video), even though it cannot be rendered using the current toolset.
> We replaced “slightly more detail” in the video with “on par”.
> Leveraging an aggressive learning rate schedule, auto-decoding converges within 200 iterations or approximately 1 second for a batch of 300 shapes. We will add this to the paper.
>
> **Minor clarifications:**
> Single-shot in table 1: Yes, this is auto-decoding with a single view. We will make this clear in the final version.
> We will make all the other requested edits to the paper.
>
> Again, thank you for your comments, and we hope that this addresses your concerns!
>
>
> [1] https://www.youtube.com/watch?v=hMx9nzG50xQ CO3D, CVPR2021: ExtremeWorkshop - David Novotny. Paper to appear in ICCV.

---

### Official Review · Reviewer_8NSa · 2021-07-07

**Rating:** 7
**Confidence:** 5

**Summary:**

The paper suggests a new representation of the 4D light field using a neural scene representation that predicts the light field from the 6D Plucker coordinates. The new scene representation enables the pixel radiance estimation using a single network query, whereas previous works needed to follow ray-marching or volume rendering procedures that require multiple queries along the ray. The presented scene representation is then utilized to learn multiple shapes or scenes using hypernetworks, in order to learn a prior of multiview consistency. Novel view synthesis results are presented for the learned scenes from the dataset, as well as generalizations from only single view supervision. Moreover, the authors demonstrate how sparse depth maps can be extracted from the learned light field in the case of Lambertian scenes.

**Limitations And Societal Impact:**

The authors did address the main limitations of the method and its potential social impacts. I believe the authors should be more apparent about extracting the (partial) depth maps only for Lambertian surfaces and that a generalization for non-Lambertian surfaces (e.g., with specular effects) is not trivial and non-obvious.

**Main Review:**

Those are the main strengths and novelty I find in the paper:
1. The paper is well written and has a good flaw. Except for Figure 3, which I find to be a bit confusing and crowded, all the figures as well the supplied video tells the story nicely and helps to understand the method concepts.
2. The idea of representing the light field using a unique coordinate system for rays is novel and interesting, but most importantly, it allows fast rendering which is currently the significant bottleneck of the existing rendering techniques. This parametrization is well motivated by the authors for 360 degrees scenes, where it is natural to assume that the radiance is constant along the ray.
3. The authors exploited the epipolar plane images properties in Lambertian surfaces and developed an equation to extract sparse depth maps from the learned light field using a single evaluation of the network and its gradient. I believe it presents a novel approach that can be leveraged in future works with neural scene representations. Nonetheless, the depth maps results are impressive given that the light field is learned without inherent representation of the geometry.

Following are the weaknesses I find in the presented method and the concerns which I would like to be addressed by the authors:
1. Although real-time rendering is a well-desired property for the task of view-synthesis, so as inherent multiview consistency, that the authors agreed that their representation does not possess. The lack of explicit 3D geometry representation makes the suggested representation not multiview consistent by design, unlike previous works, which I find to be the major limitation of the method.
2. As the authors mentioned, their representation encodes both geometry and appearance together. Hence I suspect it can mismatch between the two and compensate over geometry properties using the learned appearance. I want to emphasize that this comment differs from 1, where both limitations come from the fact the suggested representation does not model explicit geometry properties.
3. The authors suggest that a future direction for their work would be working on real data. With the above (1,2) said, it is not clear how those limitations can be addressed in the real data settings, where in those cases the need for multiview consistency is more crucial, and need to store more than one color per ray. Moreover, I'm interested to know how the representation performs on a single scene multiview reconstruction.
4. The comparison between global and local conditioning in the context of shape space learning is important and the authors describe the strengths and limitations of each option properly. However, in the context of the suggested scene representation (which is the paper's main contribution), I find the comparison to PixelNeRF irrelevant. A more proper baseline would be PixelNerF with global conditioning (meaning, auto-decoder as in the method of the paper with rendering function of NeRF). The source for that PixelNeRF overcomes LFN is not clear- is it due to the scene representation or due to the conditioning learning method? I believe there is a need to separate between those two observations to strengthen the paper's contribution. Another point of difference is that PixelNeRF utilizes positional encoding, which enables to learn higher frequencies in the light field, and I wonder why the authors did not use that.
5. In several results, it seems that thin areas are model incorrectly (smoothed or mixed), for example - figure 5 bottom chair, or the tables and benches legs in the single-shot results (10:45-55 in the supplied video). I find possible reasons for that: the LFN fails to learn the high discontinuity (frequency) of the light field where a small change in the ray change from the table leg to the background; the multiview consistency is not modeled correctly in this area; due to the low-resolution data; compensation of the learned appearance over geometry; wrong generalization of tests views. I would appreciate it if the authors could address this concern, and I suggest they show depth evaluation for the more complicated geometry areas.
6. The addition of extracting the sparse depth maps is another good contribution of the methods, however, it needs to be addressed by the authors that it is limited to Lambertian surfaces and that a generalization to secular scenes is unclear.


_________________________
After reading the other reviews and the author's response, I decided to update my rating to accept.

However, there are few points I suggest the authors clarify in the revised paper:
* In section 1.2 in the authors' response, I believe that the comparison to SDF can be misleading. Compares to LFNs, SDF is not limited to Lambertian surfaces for representing scene geometry, and the geometry representation is not partial (as the sparse depth maps extracted from LFNs).
* In section 1.4 in the authors' response, running methods like SRN and DVR with a fixed compute budget will probably yield worsened results. However, they will still be multi-view consistent, meaning that the learned scene from a novel view will correspond to the learned scene geometry.

Also, I believe incorporating all the new results presented in the rebuttal (also the new Fern results in 2.5) would serve this paper well.

Overall, I'm convinced by the novelty and contribution of this new representation, and I'm intrigued by the future works it opens up.

**Time Spent Reviewing:**

10

---

> ### Author Response · Authors · 2021-08-10
> **Response to 8NSa**
>
> We thank reviewer 8NSa for their careful review! We address the comments below  - **we regularly refer to the response overview, which is the top-level post in this forum**::
>
> > Although real-time rendering is a well-desired property for the task of view-synthesis, so inherent multiview consistency, that the authors agreed that their representation does not possess. The lack of explicit 3D geometry representation makes the suggested representation not multiview consistent by design, unlike previous works, which I find to be the major limitation of the method.
>
> Please see Sections 1.3 and 1.4 of the response overview.
>
> > As the authors mentioned, their representation encodes both geometry and appearance together. Hence I suspect it can mismatch between the two and compensate over geometry properties using the learned appearance.
>
> Please see section 1.2 of the overview for a detailed response. In general, if a 360 degree light field correctly represents multi-view appearance, then the geometry encoded in the slope of the epipolar plane images is mathematically guaranteed to be correct as well. Moreover, we validate the accuracy of the geometry by benchmarking with SRNs, and demonstrate depth map accuracy on par with SRNs; please see section 2.1.
>
> >The authors suggest that a future direction for their work would be working on real data. With the above (1,2) said, it is not clear how those limitations can be addressed in the real data settings, where in those cases the need for multiview consistency is more crucial, and need to store more than one color per ray. Moreover, I'm interested to know how the representation performs on a single scene multiview reconstruction.
>
> We note that there is existing work on extracting the geometry encoded in light-fields of complex, non-Lambertian scenes, as cited in section 1.1 of the response overview; and that for textured, Lambertian scenes, our depth-extraction methods are likely to be more effective whenever the texture can be represented, as discussed further in section 2.5. We discuss an initial experiment demonstrating that we can fit the high-frequency detail of a complex realistic scene given many observations in section 2.5. In this work we focus entirely on the few-observation regime, but we agree that exploring the single-scene, many-observation regime is an interesting direction for future work.
>
> > The addition of extracting the sparse depth maps is another good contribution of the methods, however, it needs to be addressed by the authors that it is limited to Lambertian surfaces and that a generalization to secular scenes is unclear.
>
> Please see Sec. 1.1 of the response overview.
> > The comparison between global and local conditioning in the context of shape space learning is important and the authors describe the strengths and limitations of each option properly.  … A more proper baseline would be PixelNerF with global conditioning (meaning, auto-decoder as in the method of the paper with rendering function of NeRF).
>
> There is strong evidence in the literature that NeRF-like methods with global conditioning perform poorly; we did not perform the comparison because of this evidence, and because of the computational cost of this evaluation. As one compelling bit of evidence, the authors of pixelNeRF benchmark the effectiveness of NeRF when provided with a global code in [Table 3, 1], in order to demonstrate that globally-conditioned NeRF fails in comparison to the methodology they take in that paper, in which they feed in a code derived from local features. According to [1], NeRF with global conditioning performs 2.5 db worse than LFNs on the ShapeNet chair reconstruction task. We see the systematic evaluation of LFNs with a variety of conditioning methods as an interesting avenue for future work.
>
> In addition, Facebook AI has recently benchmarked an auto-decoding, globally conditioned NeRF - see minute 18:20 of [2]. SRNs, **significantly outperform** this baseline. We will cite this result in the final version of our paper. Thank you for inquiring about this clarification!
>
> > Another point of difference is that PixelNeRF utilizes positional encoding, which enables to learn higher frequencies in the light field, and I wonder why the authors did not use that.
>
> As with any other neural implicit representation, LFNs can be combined with positional encodings, or parameterized by a SIREN network with periodic activation functions. We chose the simplest possible network architecture for this paper, as we are introducing a completely novel framework and want to focus the reader on the moving parts that are important to understand this framework. We agree that such architectural improvements are the logical next step, and will add a sentence regarding this to the discussion section.
> >In several results, it seems that thin areas are model incorrectly...I would appreciate it if the authors could address this concern, and I suggest they show depth evaluation for the more complicated geometry areas.
>
> We have performed a quantitative depth evaluation as detailed in section 2.1 of the response overview. Will add qualitative depth evaluations to the supplementary video, as described in section 2.2; please see that section for sample back-projected depth maps. We note that many chairs and other objects with thin, high-frequency details exist in our datasets which LFNs represent well, and on the chair in section 2.2, the back-projected depth map is qualitatively reasonable.
>
> >The addition of extracting the sparse depth maps is another good contribution of the methods, however, it needs to be addressed by the authors that it is limited to Lambertian surfaces and that a generalization to secular scenes is unclear.
>
> We agree, and state in our paper, that our study is limited to Lambertian surfaces, and that the generalization to specular scenes is an exciting area of future work. We note that there is past work demonstrating effective extraction of geometry information from light fields of specular scenes.  Please see Section 1.1 of the response overview for a short literature review and further discussion! We will reinforce this point in the paper, as stated in Section 1.1 of the overview.
>
> We thank you for your comments, and we hope that this clarifies our results!
>
> [1] Alex Yu, Vickie Ye, Matthew Tancik, and Angjoo Kanazawa. PixelNeRF: Neural radiance fields from one or few images. Proc. CVPR, 2020.
>
> [2] https://www.youtube.com/watch?v=hMx9nzG50xQ CO3D, CVPR2021: ExtremeWorkshop - David Novotny. Paper to appear in ICCV.

---

> > ### Comment · Reviewer_8NSa · 2021-08-16
> > **Feedback and follow-up questions**
> >
> > I thank the authors for addressing my concerns and for providing such a thorough rebuttal. I believe the additional experiment and evaluations would emphasize the contribution of the presented method. \
> > I have some follow-up questions regarding some of the experiments:
> >
> > Experiment 2.1 - how the authors compared depth estimations if LFNs can extract only partial maps?
> >
> > Experiment 2.2 - I'm impressed by those results, especially the chairs. I am just wondering whether LFNs trained on higher resolution (than the multi-class experiments)?
> >
> > Experiment 2.4 - training also done on 5 views? Or only inference? A more general question is whether training needs a full observation of the object to learn view consistency?
> >
> > Experiment 2.5 - Is it possible for the authors to supply some qualitative or quantitative results regarding the fern experiment? It is unclear how LFNs handle this scene in terms of image reconstruction (overfitting) and multi-view consistency (novel view synthesis).

---

> > > ### Author Response · Authors · 2021-08-16
> > > **Answers to follow-up questions**
> > >
> > > Thank you for your follow-up questions!
> > >
> > > > Experiment 2.1 - how the authors compared depth estimations if LFNs can extract only partial maps?
> > >
> > > As requested by NEE3 and 9CQx, we compare the depth only at valid depth estimates, i.e., only at the pixels where LFNs yield a depth estimate.
> > >
> > > > Experiment 2.2 - I'm impressed by those results, especially the chairs. I am just wondering whether LFNs trained on higher resolution (than the multi-class experiments)?
> > >
> > > Yes, these are the single-class models, which were trained at a resolution of 128x128 (see line 275 in the main paper). The NMR dataset from the multi-class experiments is only available at a resolution of 64x64, and as a result, the resolution does not suffice to infer fine detail of legs of chairs etc. which regularly are less than 5 pixels wide.
> > >
> > > > Experiment 2.4 - training also done on 5 views? Or only inference?
> > >
> > > In this case, training and inference are identical, as the training and testing objects are the same. This is following NEE3's request for an ablation that investigates the multi-view consistency as a function of the number of training views.
> > >
> > > > A more general question is whether training needs a full observation of the object to learn view consistency?
> > >
> > > The light field does not have to be observed fully at training time in order to learn multi-view consistency: Indeed, at 5 views, the light field is not fully observed.
> > >
> > > > Experiment 2.5 - Is it possible for the authors to supply some qualitative or quantitative results regarding the fern experiment? It is unclear how LFNs handle this scene in terms of image reconstruction (overfitting) and multi-view consistency (novel view synthesis).
> > >
> > > We have updated Experiment 2.5 and added videos of the context views and the intermediate views. However, we highlight that we do **not** claim multi-view consistent single-scene reconstruction, which is **an orthogonal research direction** that we are currently investigating. None of the claims in our paper rest on this experiment. We are in line with recent prior and concurrent work that investigates **prior-based reconstruction**, such as Scene Representation Networks (Sitzmann et al. 2019), Neural Scene Representation and Rendering (2018), VAE-NeRF (Kosiorek et al. 2021), ObSuRF (Stelzner et al. 2021).
> > >
> > > Thank you for taking the time!

---

> > > > ### Comment · Reviewer_8NSa · 2021-09-01
> > > > **Feedback**
> > > >
> > > > After reading the other reviews and the author's response, I decided to update my rating to accept.
> > > >
> > > > However, there are few points I suggest the authors clarify in the revised paper:
> > > > * In section 1.2 in the authors' response, I believe that the comparison to SDF can be misleading. Compares to LFNs, SDF is not limited to Lambertian surfaces for representing scene geometry, and the geometry representation is not partial (as the sparse depth maps extracted from LFNs).
> > > > * In section 1.4 in the authors' response, running methods like SRN and DVR with a fixed compute budget will probably yield worsened results. However, they will still be multi-view consistent, meaning that the learned scene from a novel view will correspond to the learned scene geometry.
> > > >
> > > > Also, I believe incorporating all the new results presented in the rebuttal (also the new Fern results in 2.5) would serve this paper well.
> > > >
> > > > Overall, I'm convinced by the novelty and contribution of this new representation, and I'm intrigued by the future works it opens up.

---

### Official Review · Reviewer_Dh8e · 2021-07-16

**Rating:** 7
**Confidence:** 4

**Summary:**

The authors propose a novel neural scene representation called Light Field Networks (LFNs), where geometry and appearance of the considered scene are represented in a 360-degree, 4D light field that is parameterized via a neural implicit representation. In contrast to other ray-marching-based or volumetric-rendering-based techniques that rely on hundreds of evaluations per ray in similar tasks, the proposed LFN only requires a single evaluation per ray, thereby significantly improving efficiency and enabling real-time rendering at low memory requirements. Key aspects to achieve this are the parametrization of the space of light rays based on Plücker coordinates. In addition, the authors embed LFNs in a meta-learning framework to allow novel view synthesis from solely sparse 2D image supervision.

The overall approach seems novel and interesting. While the complexity benefits have been demonstrated by the authors, the current approach seems to be limited to simple scenes.



**Limitations And Societal Impact:**

Limitations and societal impacts have been adequately addressed.

**Main Review:**

Originality
The approach seems novel and reasonable. In particular, the complexity benefits have also been demonstrated by the authors.


Evaluation
- In the paragraph on 'real-time rendering and storage cost' the statement might be misleading. Only references 3 and 6 have been included in Table 2, which might confuse less experienced readers. In this comparison, the authors provide an evaluation of the computational complexity w.r.t. to SRNs and pixelNeRF, where the proposed approach shows clear benefits. However, insights on the timings for different image resolutions have not been provided.
- The authors provide quantitative and qualitative results for single-shot multi-class reconstruction and class-specific single-shot reconstruction. However, the comparison only includes DVR and SRNs which seem to not be state-of-the-art approaches any more.
The qualitative comparison to further scene-overfitting approaches (i,e. NeRF-like approaches) beyond Figure 6 would be interesting as well.
- For the evaluation of global vs. local conditioning, the authors provide only a comparison to the local conditioning by pixelNeRF, that results in better quality.
- Another interesting aspect for the evaluation would be the discussion regarding which configurations of cameras can be handled, i.e. how many views are required and how the views may be distributed for a reliable scene representation.
- Limitations have been discussed.


Exposition
- The paper is well-structured and easy to follow. Figures/tables and captions are informative.
- The approach is well-motivated.
- There are a few typos that can be solved in a proof-reading. In equation 4 and the nearby text, the ray seems to be denoted inconsistently by r and l.
- In Section 5, as I understood, the reference in the paragraph 'class-specific single-view reconstruction' should read Figure 5 (instead of Figure 4).
- There is an unfinished sentence at the end of the caption of Figure 5 on page 8 and at the end of the caption of Figure 6 on page 9.


Reproducibility
- The paper seems reproducible from the facts in the paper. In addition, the authors mentioned to release code upon acceptance.

Post-Rebuttal:
I thank the authors for provding a comprehensive feedback to the reviewer's comments and agree with the and agree with the other reviewers that this significantly improves the paper. I like the presented contributions and their potential for future developments and am looking for the inclusion of the insights provided in the rebuttal into the paper and supplemental. This also makes me increase my rating towards accept.


**Time Spent Reviewing:**

4.5

---

> ### Author Response · Authors · 2021-08-10
> **Response to Dh8e**
>
> We thank the Reviewer Dh8e for the helpful review. We will address all the expository comments in the final version of the paper. Below, we hope to clarify some points  - **we regularly refer to the response overview, which is the top-level post in this forum**::
>
> > Only references 3 and 6 have been included in Table 2, which might confuse less experienced readers.
> >However, insights on the timings for different image resolutions have not been provided.
>
> The rendering speed of various 3d-structured scene representations has recently been evaluated systematically, see 17:00 of [1].  IDR and DVR are both two orders of magnitude slower than SRN, and thus of speeds the same order of magnitude as (pixel)NeRF. Please find the rendering speeds from [1] for a resolution of 800x800, with a comparison to LFNs at this new resolution, below.
>
> | Resolution       | LFN   | SRN  | DVR |  IDR |   NeRF
> | :---                   |  :----: |   :----:  | :----:  |  :----: |  :----: |
> |  800 x 800 px  |  20.5ms  |   1 s    |  ~90s   |  100s  |  ~20s |
>
> We note that the rendering speed generally is quadratic in the sidelength of the rendered image. We will add these comparisons into the paper. Thank you for the suggestion!
>
> >The qualitative comparison to further scene-overfitting approaches (i,e. NeRF-like approaches) beyond Figure 6 would be interesting as well.
>
> We agree that such a comparison would be interesting! However, in this paper, we are focused on the prior-based reconstruction regime, which has challenges fundamentally different from the overfitting regime. We discuss this distinction more fully in section 2.5 of the response overview. We see the use of LFNs in the overfitting regime as an exciting topic for a subsequent project!
>
> > However, the comparison only includes DVR and SRNs which seem to not be state-of-the-art approaches any more.
>
> To the best of our knowledge, for few-shot reconstruction in global conditioning, DVR and SRNs remain state-of-the-art. If the reviewer knows of other models that make the same assumptions, we would appreciate if they provided the appropriate references. Our claim is further highlighted by recent efforts by Facebook AI, who have benchmarked these models on their newly released CO3D dataset (see 18:30 of [1]). Please also see the previous point.
>
> >For the evaluation of global vs. local conditioning, the authors provide only a comparison to the local conditioning by pixelNeRF, that results in better quality.
>
> We did not report a comparison with globally conditioned NeRF because the poor performance of that method is part of the motivation given by the authors of pixelNeRF for introducing local conditioning, see Table 3 of the pixelNeRF paper, which shows that our method outperforms globally-conditioned NeRF on the chairs dataset by 2.5 db. Properly training volumetric rendering with auto-decoding global conditioning is further prohibitively expensive.  PixelNeRF was the only published method that performed locally-conditioned scene reconstruction from a single observation, which is why we added a comparison to PixelNeRF for completeness.
>
> Further, Facebook AI has recently benchmarked an auto-decoding, globally conditioned NeRF - see minute 18:20 of [1]. SRNs **significantly outperform** this baseline. We will cite this result in the final version of our paper. Thank you for inquiring about this clarification!
>
> >Another interesting aspect for the evaluation would be the discussion regarding which configurations of cameras can be handled, i.e. how many views are required and how the views may be distributed for a reliable scene representation.
>
> LFNs scale naturally to an arbitrary number of cameras; our paper focuses on single-shot reconstruction, but as part of the second experiment described in section 2.4 of the response overview (which evaluates multi-view consistency properties of LFNs), we evaluated LFNs with 5 camera views.
>
> Thank you for your comments!
>
> [1] https://www.youtube.com/watch?v=hMx9nzG50xQ CO3D, CVPR2021: ExtremeWorkshop - David Novotny. Paper to appear in ICCV.

---

> ### Author Response · Authors · 2021-09-01
> **Questions remaining?**
>
> Dear R Dh8e,
>
> Do you have any remaining questions or concerns following our response? Please let us know. We’d be very happy to do anything we can that would be helpful in the time remaining!

---

### Official Review · Reviewer_9CQx · 2021-07-20

**Rating:** 8
**Confidence:** 4

**Summary:**

The paper presents a new neural scene representation based on the idea of light fields. Rather than predicting properties (e.g. occupancies, colors) for points in space, the paper proposes to predict such entities for all rays in a scene using simple MLP networks.
For the network input the authors propose to use Plucker coordinates to canonically parametrize the viewing rays independently of a point offset.

To encourage multi-view consistency of the network predictions, the authors propose a meta-learning approach allows to decouple the rendering from the latent code optimization.

In contrast to volumetric neural scene representations that require expensive sampling with multiple network predictions per ray for rendering a novel view, the proposed method only requires a single network evaluation per ray while achieving state-of-the-art rendering results.



**Ethical Concerns:**

There are no ethical concerns.

**Limitations And Societal Impact:**

The limitations and societal impact have been sufficiently discussed.



**Main Review:**

Paper Strengths:

#Originality
The paper presents several novel ideas for new ways of encoding neural scene representations and demonstrates their viability.
Neural scene representations are an important and vibrant research direction and this paper contributes nicely by introducing a new model which tackles common scalability issues although it also introduces new limitations.

#Quality
The paper is well structured and written. Illustrative figures support the explanations in the text well. The paper also presents competitive state-of-the-art results.

#Clarity
The paper is clearly written and the mathematical model description is sound (apart from a few small mistakes - see comments below).

#Significance
I believe the paper contains several valuable ideas and thoughts that should be shared with the community and hence merit publication.




Paper weaknesses / questions:

1. The experimental evaluation could be stronger in my opinion.

1.1. Ablations: One can certainly assume that the method will not work without the proposed meta-learning network architecture, but it is not explicitly stated or empirically shown that/why a simpler network architecture would not work.
Moreover, it is unclear how changes of the latent code size or the network size would affect the output quality, as well as overfitting vs. generalization properties.

1.2. Stronger evaluations: One of the major paper claims is that LFNs are able to encode both the geometry and the appearance of a scene (L66). The evaluation of these two entities is not very strong.

1.2.1 For the appearance, all scenes contain only simple piece-wise constant textures with barely any challenging high-frequency details. Since, the network sometimes already struggles to recover these simple scene sharply, one can assume that more complex textures cannot be well recovered.
Thus, the paper does not really show that LFNs are able to encode real-world textures. It would still be nice to see some results on real images.
Further, it would be interesting to know/discuss whether the appearance modeling quality could be improved with an increased network capacity / larger latent codes.

1.2.2 For the geometry, there is no quantitative evaluation at all.
Although the depth values are only sparsely recovered, it is still possible to compute error values for corresponding masked depth maps. Otherwise the quality of computed depth maps (Eq. (8)) is difficult to assess.
Besides showing depth maps, one could also (sparsely) evaluate the quality of surface normals to better assess the geometric reconstruction quality.




Minor comments:

- several parameters have not been specified in the paper:
  what is the latent vector size k (Eq. (9))?
  what is the LFN parameter size \ell (Eq. (9))?
  what is the value of \lambda_lat in Eqs. (10), (11) ?

Although some parameters are described in the supp. mat., it does not need much space to state their values directly in the paper to make it better self-contained and give the reader a feeling for the network and optimization parameters.
\lambda_lat is only called \lambda in the supp. mat.


- Fig. 2 is very helpful. Although, the figure illustrates several complex concepts which are detailed in the text below one could improve the caption to make the figure more self-contained and briefly explain the symbols as many of them are not explained in the caption.

- L194: it is better to use a different symbol than \ell to denote a ray since the same symbol is used to denote the number of LFN parameters in Eq. (6) / L154.

- L202.5: This should probably also be a numbered equation. Even if you do not reference it, other might want to.
There is a slight error in there: The right hand side of the “element of” operator describes the space of a tuple, while the left hand side is not a tuple, but a set of tuples.

- L219: “which sends the” -> “which maps the” ?

- Eq. (10): z_j -> z_i ? (z_j does not seem to make sense here)

- Table 1: best numbers are incorrectly highlighted in 2nd and 3rd last columns


--------------------------------------------------------------------------------------------------------------
#Post-Rebuttal:
I agree with the other reviewers that the authors did an excellent job in responding to the reviewer concerns.
Thanks a lot for this great effort!
Overall, I am very happy with the responses not only to my questions, but also to the ones of the other reviewers.
Therefore, I will keep my positive rating to accept the paper as I believe it has a lot of valuable insights that should be shared with the community and which merit publication. Thanks for the great work!


**Time Spent Reviewing:**

8

---

> ### Author Response · Authors · 2021-08-10
> **Response to 9CQx**
>
> We thank the Reviewer 9CQx for the detailed and constructive review. We will perform all the changes requested in the minor comments. We first answer one of the minor comments  - **we regularly refer to the response overview, which is the top-level post in this forum**:
>
> > several parameters have not been specified in the paper: what is the latent vector size k (Eq. (9))? what is the LFN parameter size \ell (Eq. (9))? what is the value of \lambda_lat in Eqs. (10), (11) ?
>
>
> In our experiments, K =256,  Ell= ~400,000 , lambda_lat = 1. We thank the reviewer for asking and we will add these numbers to the paper.
>
> We now address the major points of discussion:
> > 1.1. Ablations: One can certainly assume that the method will not work without the proposed meta-learning network architecture, but it is not explicitly stated or empirically shown that/why a simpler network architecture would not work.
>
> We agree that it is instructive to benchmark the proposed hypernetwork-based approach with a simpler conditioning-via-concatenation based approach. Please see section 2.3 of the response overview for our results, which show that conditioning-via-concatenation performs significantly worse than our hypernetwork approach.
>
> However, we ask that the reviewer please note that we do **not** claim that the proposed method outperforms other forms of conditioning, such as gradient-based meta-learning or FILM conditioning. We will clarify this in the meta-learning section and add the ablation study we have performed. Systematically evaluating LFNs in the context of more sophisticated forms of conditioning is an interesting avenue for future work!
>
> >  Moreover, it is unclear how changes of the latent code size or the network size would affect the output quality,  …
>
> We agree that running a formal study here would be an experiment of great interest, but unfortunately, it is not computationally feasible for our research group to investigate all hyperparameter dimensions of the proposed model. We chose latent code size and network size such that they would be comparable to NeRF, SRNs, DVR, IDR, DeepSDF, IM-Net, OccupancyNetworks, etc. Informally, we can state that in our experiments in this and earlier projects on meta-learning neural implicit representations, output quality is fairly robust to latent code sizes between 128 and 512, as well as network depths of 4 to 8 and widths of 256 and 512 for hypernetwork-based conditioning, while performance degrades for less than 4 layers, latent codes smaller than 128, or fewer than 256 hidden units.
> > … as well as overfitting vs. generalization properties.
>
> While we do not investigate the effects of the latent code size or network size on these properties, we have added an ablation study investigating the number of objects in the training set vs. the reconstruction performance and multi-view consistency - please see Section 2.4 of response overview!
>
> > 1.2.1 For the appearance, all scenes contain only simple piece-wise constant textures with barely any challenging high-frequency details. Since, the network sometimes already struggles to recover these simple scene sharply, one can assume that more complex textures cannot be well recovered. Thus, the paper does not really show that LFNs are able to encode real-world textures. It would still be nice to see some results on real images.
>
> While we do not solve the (wide-open) problem of inferring a compact representation of a scene containing high-frequency textures given incomplete observations of the scene, we have run an experiment demonstrating that LFNs can capture realistic, high-frequency details when allowed to overfit on a single scene. Please see section 2.5 of the response overview for a detailed discussion of this point.
>
> > 1.2.2 For the geometry, there is no quantitative evaluation at all. Although the depth values are only sparsely recovered, it is still possible to compute error values for corresponding masked depth maps.
>
> We thank the reviewer for this suggestion, and we have **performed quantitative and qualitative evaluations of depth extraction**. Please see Sec. 2.1 and 2.2 of the response overview. We found that in our quantitative evaluation, we outperformed SRN on valid depth estimates. We will add these results to the paper.
> Thank you for your comments!

---

### Author Response · Authors · 2021-08-10
**Author Response - Additional Experiments [2/2]**

## 2. Additional Experiments

###  2.1. Quantitative depth evaluation.
We benchmark mean L1 error on valid depth estimates with Scene Representation Networks (SRNs) in the setting of single-shot single-class reconstruction. SRNs have recently been confirmed as a strong baseline for depth estimation on Facebook AI’s newly released CO3D dataset, offering better depth maps than DVR [1]. We measure mean absolute distance on foreground depth values. Following are the quantitative results:

| Method      | Cars  | Chairs |
| ----------- | ----------- | ----------- |
| SRNs     | 0.071       | 0.20      |
| LFNs   |   0.059        | 0.130       |

Surprisingly, LFNs outperform SRNs on both cars and chairs. We note that LFNs have a slight advantage in this comparison, as we can only benchmark on the sparse depth values, for which LFNs have high confidence. This includes occlusion boundaries, which are areas where the sphere-tracing based SRN incurs high error, as it is forced to take smaller and smaller steps and may not reach the surface (see section 1.4).

Nevertheless, this is conclusive evidence for our claim that the proposed algorithm can extract valid depth from an LFN. Still, as highlighted in Sec. 1.2, we will highlight that we do not claim that this algorithm is competitive for 3D reconstruction - in particular, we will **not** claim that LFNs perform better geometry reconstruction than SRNs, DVR, IDR, NeRF, etc., highlighting that LFNs only extract sparse depth maps. We will instead merely offer this as empirical evidence of the claim that LFNs implicitly encode depth information, that it is **in principle** possible to extract it, and highlight that it is a great direction of future work to conceive algorithms that may be used to extract dense 3D geometry from LFNs.

###  2.2. Qualitative depth evaluation.
As suggested by R 9CQX, R NEE3, R 8NSA, we will add examples of the point clouds resulting from back-projecting sparse depth maps into 3D to the supplemental video. Please see the following fully anonymous links for (random, non cherry-picked) **preliminary results**:  **[cars](https://streamable.com/f46hjy)** and **[chairs](https://streamable.com/xjmf3y)**. These point clouds were generated by 4 novel views of reconstructions from the single-class experiments. As the video demonstrates, the point clouds extracted with the proposed algorithm clearly delineate the shape of the underlying 3D scenes, **including in areas of fine geometry, as requested by R 8NSA**. We will further add an analysis of failure cases due to multi-view inconsistency to the supplementary video. Consistent with the quantitative evaluation, the qualitative depth evaluation clearly supports our claim that LFNs encode shape information. Following our clarification above, we will further add a section to the video that explains the relationship between epipolar lines extracted from LFNs and the depth, basically an animated version of Fig. 1 in the supplemental material.

###  2.3. Hypernetwork vs. conditioning via concatenation / ablating simpler network architecture.
R 9CQX and R NEE3 remark that an ablation study with a simpler network architecture / a concatenation-based meta-learning framework would offer further insight. We agree, and have run an experiment to compare hypernetworks to the simpler alternative of conditioning via concatenation. We parameterize the LFN identically to the MLPs in pixelNeRF and DVR: A 5-layer, fully-connected residual MLP with ReLU activations and 512 hidden units. The latent code is directly concatenated to the input ray coordinate. Experiment settings are identical to “Multi-class Single-View Reconstruction” (line 266). Both methods were trained for approx. 3 days on a single RTX 6000 GPU. In this setting, conditioning via concatenation yields significantly worse performance:


| Method      | plane |  bench | cbnt. | car |  chair | disp. | lamp |  spkr. | rifle |  sofa |  table |  phone | boat | mean |
| -----------      | ----------- |  ----------- | ----------- | ----------- |  ----------- | ----------- | ----------- |  ----------- | ----------- |  ----------- |  ----------- |  ----------- | ----------- | ----------- |
| Hypernetwork      | 29.95 | 23.21 | 25.91 | 28.04 | 22.94 | 20.64 | 24.56 | 22.54 | 27.50 | 25.15 | 24.58 | 22.21 | 27.16 | 24.95 |
| Conditioning via Concatenation | 24.08   |  19.43  | 21.01 | 23.57 | 19.81 | 17.47 | 19.05 | 19.40 | 22.24 | 20.85  | 19.81 | 17.64 | 24.02 | 20.64|

All results are PSNR in dB. This is in line with the insight from Pi-GAN (Chan et al., CVPR 2021), where FILM-conditioning, which is an intermediate between a hypernetwork and conditioning via concatenation, was found to perform significantly better than conditioning via concatenation. Similarly, Facebook AI has found the hypernetwork-conditioned SRN to outperform concatenation-conditioned DVR and NeRF alternatives [1]. We further note that conditioning via concatenation is a special case of a hypernetwork, namely a single linear layer outputting the biases of the first LFN layer - see for instance appendix of MetaSDF, Sitzmann et al. 2020 for a proof.


We note that we do **not** claim that the proposed method of conditioning is superior to any other method of conditioning. Rather, the proposed LFNs framework is compatible with **any** conditioning method. We similarly do **not** claim that the auto-decoder framework is superior to other inference methods (see line 224). Neither of these aspects are central to the LFN framework - we merely argue that to learn multi-view consistent light fields, we require **some form** of meta-learning. We will clarify this in the meta-learning section. We will further highlight in the discussion that the means of conditioning and inference are natural directions for future study. Lastly, we will add this ablation study to the paper and we will include the conditioning-via-concatenation variant in the published code, in order to facilitate further work in this direction.

### 2.4. Multi-view consistency vs. training set size / ablating overfitting vs. generalization.
We have run the exact experiment that the reviewer recommends: we benchmark PSNR of novel views on 10 objects, reconstructed from a single observation using models trained on 1, 10, 100, 1k, all shapes. For the multi-view setting, we take this as counts per-class (i.e., 1 shape per class, 10 shapes per class, etc, as well as a constant 10 shapes per class in the test set). We report novel view synthesis quality in PSNR (dB) in the following table:

| Dataset  | 1  | 10 | 100 | 1k | all |
| ----------- | ----------- | ----------- | ----------- | ----------- |----------- |
| Cars (Single-class)   |   16.21        | 18.98       | 20.76 | 23.09 | 23.56 (~2.5k objects) |
| NMR (Multi-Class)   |   18.03        | 19.77       | 20.82 | 23.87 |  24.95 (~2k objects per class) |

Where the first column denotes the dataset & experiment setting (Single-class vs. multi-class), and the other columns denote the number of object instances in the training set. Performance increases monotonically with the number of objects in the training set. Additional objects would likely improve performance further. This is evidence for the claim that meta-learning on a dataset of 3D shapes enables consistent novel view synthesis on test objects.

However, we note that meta-learning across a dataset of 3D scenes serves two distinct roles: (1) learning a space of multi-view consistent light fields (2) learning a prior over 3D scenes to enable few-shot reconstruction. To better disentangle these two roles, we ran a second experiment. We focus on the single chair class. For each instance in the training set, we randomly pick 5 views, such that on average, almost every surface point of each chair has been observed, but **not every oriented ray**. I.e., while the 3D surfaces of the chairs are densely sampled, their light fields are not. We now train LFNs on 10, 100, 1000, and all (4.5k) chairs, **including the 10 chairs previously mentioned**, and evaluate view synthesis performance in PSNR on a held-out set of novel views for each of these 10 chairs. Therefore, we test only property (1) of the meta-learning approach - i.e., how the meta-learning learns multi-view consistency as an abstract property - without entangling it with the capability to perform few-shot reconstruction. Please find the quantitative results below:

|    | 10 | 100 | 1k | all
| ----------- | ----------- | ----------- | ----------- | ----------- |
| PSNR (dB) on novel views  |   15.88        | 16.83       | 19.12 |  22.15 (~4.6k objects) |

Where the columns denote the number of instances in the training set. Again, PSNR improves monotonically with the number of training objects. This provides further, strong evidence that the meta-learning approach enables LFNs to perform multi-view consistent view synthesis, and that more training objects might lead to further gains.

In summary, the results suggest that multi-view consistency improves monotonically as a function of the number of training objects. We thank the reviewers for this suggestion, which highlights an exciting direction for future work. We will include this experiment together with a discussion into the paper, and add qualitative results into the video.

---

> ### Author Response · Authors · 2021-08-10
> **Author Response - Additional Experiments [2/2] - continued**
>
> ###  2.5. Real-world Scenes & Textures.
> Reviewer R 9CQx has asked whether LFNs can represent high-frequency real-world textures, while reviewer R 9NSa has asked whether LFNs can represent high-frequency detail in light-space. We have run an experiment to validate this point; however, we must first make a clarification. We can distinguish between two regimes.
>
> 1. The overfitting regime, where a neural scene representation is fit to a 3D scene that is **completely observed**, i.e., every surface point is observed enough times to enable triangulation.
> 2. The prior-based reconstruction regime, where we aim to reconstruct a 3D scene given **incomplete observations**.
>
> Please note that problem (2) is of great interest in the field of artificial intelligence, where we regularly infer neural scene representations from incomplete observations. LFNs are immediately applicable to this regime, opening avenues of future work in scene understanding, scene decomposition, reinforcement learning (where a tractable inverse graphics model is of critical importance), etc.
>
> In this paper, we follow a large body of prior and concurrent work and investigate (2). In this regime, __it is an outstanding open problem to find any global-conditioning method that can recover high-frequency detail__. Such a method does not exist to date. Local conditioning (i.e., PixelNeRF-like approaches) can sometimes recover high-frequency appearance, though even in that case, subsequent overfitting is usually required for high visual quality.
>
> We agree that problem (1) is similarly of great interest, mainly for applications in computer graphics. In the non-neural setting, prior work has demonstrated novel view synthesis with light fields in complex real world scenes in this regime (e.g. [38] in paper). However, this regime requires a different set of methods - we are pursuing this in follow-up work, but believe that this exceeds the scope of this paper.
>
> Yet, we agree that a proof-of-concept that LFNs **can in principle represent high-frequency information** is useful to motivate future work in this direction. We have run an experiment to validate this: We fit an LFN to all the views of the “Fern” scene from the NeRF dataset, which consists of photos of a real-world scene captured with a DSLR camera. We parameterize the LFN as a SIREN (Sitzmann et al. 2020). LFNs succeed in reproducing all context images perfectly, proving that in principle, an LFN is capable of parameterizing realistic, 4d high-frequency content. Please see **[this video](https://streamable.com/iusq5y)** for the context views that have been perfectly overfit with LFNs. As expected, rendering out the intermediate views leads to basically random images, please see **[this video](https://streamable.com/qqejsp)**, as there is no mechanism to enforce multi-view consistency, in contrast to the experiments in the paper, where this prior was learnt using meta-learning. We are currently investigating this interesting direction, but note that this is outside the scope of this paper, which does not make any claims on overfitting single scenes.
>
> We will add this discussion, together with the additional experiment, to the paper.  We leave the systematic validation of LFNs in the context of complex real-world scenes to a subsequent paper. We see this as one of the most exciting new directions opened up by this work.
>
>
> # Conclusion
> To summarize, we thank the reviewers for their careful feedback and additional suggestions for evaluation, which will make the paper significantly stronger. We look forward to further discussion, and are happy to answer any questions that might arise.
>
> [1] see minute 16:30 and following in https://www.youtube.com/watch?v=hMx9nzG50xQ CO3D, CVPR2021: ExtremeWorkshop - David Novotny. Paper to appear in ICCV.
>
> [2] Depth Estimation for Glossy Surfaces with Light-Field Cameras (Tao et al., ECCV 2014)
>
> [3] Depth Estimation and Specular Removal for Glossy Surfaces Using Point and Line Consistency with Light-Field Cameras (Tao et al., PAMI 2016)
>
> [4] SVBRDF-Invariant Shape and Reflectance Estimation From Light-Field Cameras (Wang et al.)

---

### Author Response · Authors · 2021-08-10
**Author Response - Clarifications [1/2]**

We thank the reviewers for their careful reading, and detailed and considerate feedback.

We are glad that reviewers unanimously agree that Light Field Networks (LFNs) are a novel and well-motivated neural scene representation (“several novel ideas” (9CQx), “novel and reasonable” (Dh8e), “novel and interesting” (8NSa), “novel and interesting” (NEE3)). We’re further glad that reviewers agree that this direction is impactful and that its key aspect - single-evaluation rendering and its complexity - is clearly validated (“tackles common scalability issues” (9CQx), “the complexity benefits have also been demonstrated” (Dh8e)), and that the paper is well written (“well written” (NEE3), “well written and good flow” (8NSa), “well-structured and easy to follow” (Dh8e), “well structured and written” (9CQx)).

The reviewers also agree that further ablations and empirical results will make the paper stronger, serving to highlight its strengths, clarify limitations, and outline important directions for future work.  **We agree, and are happy to report that we were able to execute the majority of the suggested experiments - see Sec. 2 as well as per-reviewer responses.** First, however, we would like to offer several clarifications.

## 1. General Clarifications
### 1.1. Lambertian scenes
R 8NSa is correct: the proposed method for extracting depth from light fields assumes Lambertian scenes and will not succeed otherwise. While we mention this assumption (lines 157, 178, 311), we will explicitly state in the limitations section and in the paragraph “Extracting depth maps from LFNs” that the proposed algorithm will not work otherwise, that it is not clear how to extend it to specular scenes, and that this is an important direction for subsequent research. We will further point to prior non-neural-network-based studies in this direction [2-4].

This paper aims to introduce a novel scene representation with fundamentally different trade-offs from existing 3D-structured neural scene representations. We focused on Lambertian scenes in order to keep the paper simple and focused on this new perspective. We see the application of LFNs to non-Lambertian scenes as an exciting avenue for future work. In the rest of our response, we will assume unless explicitly stated otherwise that we are referring only to Lambertian scenes.

###  1.2. Light fields encode depth
We agree that qualitative and quantitative evaluation of depth will make the paper stronger. We have run these experiments and will discuss them as below in Sec. 2. However, the reviewers’ comments have guided our attention to an important nuance that we’d like to clarify here. In particular, we’d like to disentangle two points:

1. An LFN that correctly maps all rays to their colors __necessarily__ encodes the geometry of the underlying scene via the slope of the lines in the epipolar plane image: LFNs are an implicit representation of scene geometry.
2. There exists a method by which we may efficiently measure the slope of the epipolar lines, and can therefore efficiently extract the geometry encoded in the LFN.

In particular, (1) is a fundamental property of **any** 360 degree light field. If an LFN correctly reproduces the colors of all rays, the slopes of the Epipolar Plane Images (EPIs) (Eq. 7, Fig. 3 (top left) and (bottom left), inline figure in line 188) **correlate exactly** with the depth of the respective rays. This relationship is also apparent in Eq. 8, where the gradients are an estimate of the slope of the EPI lines. This gives an explicit relationship of the slope of the EPI lines to the depth of the respective ray. The derivation / proof of this statement is found in the appendix.

Fig. 2. (center) and (right), Fig. 3, (top left) and (bottom left), the figure inset in line 188, as well as the figure in the supplemental material visualize this fact. If the multi-view appearance of the light field is correct, then the EPIs are correct. If the EPIs are correct, then the geometry, encoded in the slope of the epipolar lines, is also correct.

For all results presented in the video that are multi-view consistent (which is a significant majority), it follows that their geometry is necessarily correctly encoded in the geometry of their 4D LFN. This addresses the questions raised by R 8NSa: whenever appearance is incorrect, the geometry will similarly be inaccurate. But if the multi-view appearance is correct, the geometry is correct as well. Further, in the case of multi-view inconsistency, geometry is only incorrect in the parts of the light field that are multi-view inconsistent.

A 360 degree light field can thus be seen as an implicit representation of scene geometry. This point is **independent** of whether or not the geometry can be extracted easily, which is point (2). Like in an SDF, where the geometry is implicitly encoded in the zero-level set and has to be extracted via cube-marching or ray-marching, the proposed algorithm is only one method to extract partial information about the geometry implicitly encoded in an LFN.

We do **not**  claim that the algorithm detailed in the paper is a competitive method for explicit 3D reconstruction. Instead, our sparse depth map extraction is only meant to give empirical evidence that LFNs indeed contain the geometry of the underlying scene. We will clarify this in the paper and will rephrase claim 4 in line 64 as:
> We demonstrate that inferred LFNs encode both appearance and geometry of the underlying 3D scenes by extracting sparse depth maps from the derivatives of LFNs, leveraging their analytical differentiability.

For our simple algorithm based on local derivatives, we demonstrate in our experiments (see 2.1, 2.2. below) that the extracted sparse depth maps are competitive with the depth extracted by SRNs. We note that significantly more advanced methods for depth extraction exist (such as line detection via Hough-voting in the Epipolar Plane Image) that will further improve results. We see the problem of developing methods for extracting meshes and occupancy functions from LFNs as a fascinating area of future research opened up by the introduction of this novel neural scene representation. A systematic study adapting previous light-field geometry extraction methods unfortunately exceeds the scope of this work, although we hope to continue this investigation going forward.

We thank the reviewers for helping us to clarify this point. We will incorporate this discussion, clearly delineating between statements (1) and (2), and highlighting that we do not claim that LFNs can readily be used for full geometry reconstruction,  into the discussion section and the depth-extraction section of the paper.

Please see section 2 below for further qualitative and quantitative depth evaluations of LFNs.

### 1.3. Multi-view consistency in practice
We strongly agree with R 8NSa and R NEE3 that the fact that LFNs do not strictly enforce multi-view consistency is a limitation, and a neural network architecture that strictly enforces multi-view consistency is an exciting avenue for future work. We will highlight this in the limitations section as well as the discussion. However, we note that LFNs are in practice **predominantly** multi-view consistent on the datasets that we consider. This is evident from the video results (7:54 and following), which are not cherry-picked. Only when reconstruction itself fails are LFNs not multi-view consistent. We further note that we generally outperform both SRNs and DVR, which, as noted by R NEE3, is (albeit not strict) evidence for multi-view consistency, as PSNR and SSIM penalize deviation from correct RGB colors. Further, LFNs are almost perfectly multi-view consistent on all scenes in the training set. In Sec. 2, we provide further strong empirical evidence that meta-learning enables inference of multi-view consistent LFNs.

### 1.4. Multi-view consistency in principle
R 8NSa argues that it is a major weakness of LFNs compared to SRNs, DVR, NeRF, pixelNeRF, etc. that they do not strictly enforce multi-view consistency. We agree that ray-marching is an attractive way to enforce multi-view consistency **in the case of bounded scenes**. However, we note that given a fixed compute budget, any ray-marching based method can in practice be forced to be multi-view inconsistent. Given a fixed number of steps, sphere-tracing in SRNs or DVR will fail to converge to the surface of scenes with a large depth range or fine detail, for instance in long corridors, grazing objects at their silhouette, or through the leaves of bushes, resulting in multi-view inconsistency. Similarly, the depth range or complexity of a scene can be increased until volumetric rendering fails to correctly approximate the rendering integral, and we note that volumetric rendering requires a-priori knowledge of a correct near and far plane. These limitations are not theoretical and appear in practice, as showcased in Facebook AI’s large-scale study of different inverse graphics pipelines on the CO3D dataset [1]
We argue that LFNs offer an intriguing orthogonal approach to neural rendering that may in future work inform or even be combined with 3D-structured neural representations. We will highlight this issue in the paper and include this discussion.


[1] see minute 16:30 and following in https://www.youtube.com/watch?v=hMx9nzG50xQ CO3D, CVPR2021: ExtremeWorkshop - David Novotny. Paper to appear in ICCV.

[2] Depth Estimation for Glossy Surfaces with Light-Field Cameras (Tao et al., ECCV 2014)

[3] Depth Estimation and Specular Removal for Glossy Surfaces Using Point and Line Consistency with Light-Field Cameras (Tao et al., PAMI 2016)

[4] SVBRDF-Invariant Shape and Reflectance Estimation From Light-Field Cameras (Wang et al.)

---

### Decision · Program_Chairs · 2021-09-27

**Decision:**

Accept (Spotlight)

**Comment:**

The submission was thoroughly reviewed and discussed. All four reviewers support acceptance. The work was found to be stimulating and can usefully inform follow-up efforts in this area. The AC supports the reviewers' recommendation. The authors are encouraged to thoroughly address the reviewers' concerns and recommendations in the revision.